# Altered potassium channel distribution and composition in myelinated axons suppresses hyperexcitability following injury

Margarita Calvo[1,2,3]*, Natalie Richards[1], Annina B Schmid[4,5†], Alejandro Barroso[1,6†], Lan Zhu[1,7†], Dinka Ivulic[2], Ning Zhu[5], Philipp Anwandter[8], Manzoor A Bhat[9,10], Felipe A Court[11,12,13], Stephen B McMahon[1], David LH Bennett[5]*

[1]Wolfson Centre for Age-Related Diseases, Kings College London, London, United Kingdom; [2]Departamento de Fisiologia, Facultad de Ciencias Biologicas- Pontificia Universidad Catolica de Chile, Santiago, Chile; [3]Departamento de Anestesiologia, Facultad de Medicina, Pontificia Universidad Catolica de Chile, Santiago, Chile; [4]School of Health and Rehabilitation Sciences, The University of Queensland, Brisbane, Australia; [5]Nuffield Department of Clinical Neurosciences, University of Oxford, Oxford, United Kingdom; [6]Hospital Regional Universitario de Málaga. Servicio de Anestesiología, Málaga, Spain; [7]School of Allied Health Sciences, Faculty of Health and Life Sciences, De Montfort University, Leicester, United Kingdom; [8]Departamento Ortopedia y Traumatologia, Facultad de Medicina, Pontificia Universidad Catolica de Chile, Santiago, Chile; [9]Department of Physiology, UT Health Science Center at San Antonio, San Antonio, United States; [10]School of Medicine, UT Health Science Center at San Antonio, San Antonio, United States; [11]Center for Integrative Biology, Universidad Mayor, Santiago, Chile; [12]FONDAP, Geroscience Center for Brain Health and Metabolism, Santiago, Chile; [13]Millenium Nucleus for Regenerative Biology, Pontificia Universidad Catolica de Chile, Santiago, Chile

*For correspondence: mcalvob@ uc.cl (MC); david.bennett@ndcn. ox.ac.uk (DLHB)

†These authors contributed equally to this work

Competing interests: The authors declare that no competing interests exist.

**Abstract** Neuropathic pain following peripheral nerve injury is associated with hyperexcitability in damaged myelinated sensory axons, which begins to normalise over time. We investigated the composition and distribution of shaker-type-potassium channels (Kv1 channels) within the nodal complex of myelinated axons following injury. At the neuroma that forms after damage, expression of Kv1.1 and 1.2 (normally localised to the juxtaparanode) was markedly decreased. In contrast Kv1.4 and 1.6, which were hardly detectable in the naïve state, showed increased expression within juxtaparanodes and paranodes following injury, both in rats and humans. Within the dorsal root (a site remote from injury) we noted a redistribution of Kv1-channels towards the paranode. Blockade of Kv1 channels with α-DTX after injury reinstated hyperexcitability of A-fibre axons and enhanced mechanosensitivity. Changes in the molecular composition and distribution of axonal Kv1 channels, therefore represents a protective mechanism to suppress the hyperexcitability of myelinated sensory axons that follows nerve injury.

**eLife digest** Around 20% of the world's population experiences long-lasting "chronic" pain, which often results in poor sleep, depression and anxiety. One of the most disabling forms of chronic pain is called neuropathic pain, which results from injuries to sensory nerves. Pain or discomfort is felt in response to touches that are not normally painful. Neuropathic pain is difficult to treat as we do not fully understand the molecular mechanisms that cause it.

Stimulating a nerve causes it to produce action potentials. At a molecular level, these action potentials are generated by ions moving into and out of the neuron through proteins called ion channels. The movement of sodium ions into a neuron triggers an action potential, and the movement of potassium ions out of the neuron returns it to a resting state.

After a sensory nerve is cut or otherwise damaged it becomes hyperactive and produces spontaneous electrical activity that the brain interprets as pain signals. However, it is not fully understood how cutting a nerve affects the ion channels in a way that generates this hyperactivity.

Different types of ion channel are found in different regions of the nerve cell; for example, type 1 potassium channels are normally found in a region called the juxtaparanode at the axon of the neuron. Calvo et al. have now tracked what happens to type 1 potassium channels after nerve injury in rats. Soon after nerve damage occurred, nearly all of these ion channels disappeared from the juxtaparanode. At the same time, electrical activity in the cut nerve increased, and the recovering animals responded in ways that suggested they were hypersensitive to the nerve being touched.

Three weeks after the injury, most rats lost their hypersensitivity and the electrical activity in the cut nerve returned to near-normal levels. Calvo et al. found that the recovering nerves contained new subtypes of type 1 potassium channels. These potassium channels did not just appear in the juxtaparanode: they also invaded the 'fence' region that normally separates potassium channels from sodium channels. The same was observed to happen in the nerves of patients that suffer from neuropathic pain due to a nerve injury.

At this late time point after nerve injury, blocking the activity of potassium channels produced the same abnormal increase in the nerve's electrical activity as seen immediately after the nerve had been cut. The rats' hypersensitivity to touch also returned. This suggests that the appearance of the new potassium channel subtypes might be a protective mechanism that reduces the activity of a damaged nerve to decrease pain.

These findings suggest new ways of treating neuropathic pain. Further studies are now needed to investigate whether drugs that can activate the new potassium channel subtypes could stop pain from an injured nerve becoming a long-term problem.

## Introduction

Following traumatic nerve injury spontaneous activity develops initially in myelinated and subsequently in unmyelinated sensory axons (*Wall and Gutnick, 1974*; *Kajander and Bennett, 1992*; *Boucher et al., 2000*; *Michaelis et al., 2000*; *Wu et al., 2001*; *Liu et al., 2000a*). The onset of this spontaneous activity is associated with the emergence of pain-related sensory changes in animal models and is critical for the maintenance of peripheral neuropathic pain (*Haroutounian et al., 2014*) in patients where selective blockade suggests the involvement of myelinated axons (*Campbell et al., 1988*). Ectopic activity is particularly prominent in myelinated afferents and peaks within the first few days post injury and then declines over subsequent weeks (*Kajander and Bennett, 1992*; *Liu et al., 2000a*; *2000b*; *Han et al., 2000*). Such ectopic activity arises both at the neuroma site and also at the level of the dorsal root ganglion (DRG) (*Han et al., 2000*; *Liu et al., 2000b*; *Amir et al., 1999*; *2005*; *Wall and Devor, 1983*).

Altered expression, function and trafficking of voltage-gated ion channels are key determinants of these excitability changes. *Shaker* type voltage-gated potassium channels (Kv1 channels) are important determinants of neuronal excitability. They are formed by heteromultimers of α and β subunits (*MacKinnon, 1991*). The characteristics of the outward currents they carry depend on subunit composition. Sensory neurons are known to express Kv1 channels and functionally these channels have been shown to limit excitability of sensory neurons: For instance Kv1.2 suppresses

excitability at the level of the sensory neuron cell body (*Gold et al., 1996*; *Rasband et al., 2001*; *Zhao et al., 2013*; *Everill et al., 1998*) and Kv1.1 acts as a 'brake' on mechanosensitivity at the terminals of C-mechano-nociceptor and Aβ-mechanoreceptors (*Hao et al., 2013*). Kv1 channels also act as excitability brakes for cold thermal sensitivity in intact and damaged axons of primary sensory neurons (many of such fibres are also mechano-sensitive) (*Roza et al., 2006*; *Madrid et al., 2009*). Kv1 channels are known to be expressed in the juxtaparanodal region of myelinated sensory axons. An unexplored issue, however, is whether the distribution of these channels changes under pathological neuropathic states.

Saltatory conduction in myelinated fibres depends on the molecular organization of channel domains within the axon (*Chang and Rasband, 2013*): voltage-gated sodium channels (Nav) are clustered at the node of Ranvier. Nodes are flanked by the paranode, which is an important point of attachment between the axon and the terminal loops of the Schwann cell. Just inside the innermost axoglial junction of the paranode is the juxtaparanode a domain enriched in Kv1 channels Kv1.1 and 1.2. The localisation of Kv1.1 and 1.2 to the juxtaparanode is dependent on the formation of a molecular scaffold, which includes the adhesion molecules caspr2 and TAG-1 (*Poliak et al., 2003*). In the naïve state in adulthood, the juxtaparanodal Kv1 channels are thought not to have a major influence on axon conduction properties of peripheral myelinated axons (*Poliak et al., 2003*; *Chiu and Ritchie, 1980*; *Sherratt et al., 1980*; *Rasband et al., 1998*), probably because they are electrically insulated from the node of Ranvier under the myelin sheath. However during development (*Vabnick et al., 1999*) and following primary demyelination (*Rasband et al., 1998*) (during which myelin is removed but the axon remains intact), Kv1.1 and 1.2 become more widely distributed to include the paranode and even the node (*Arroyo et al., 2004*), and can act to suppress excitability. Although Kv1.1 and 1.2 expression within the soma is known to be down-regulated following axon transection, and this leads to hyperexcitability at the soma (*Rasband et al., 1998*; *Ishikawa et al., 1999*; *Park et al., 2003*), the distribution of these channels at the nodal complex and damaged nerve terminal (in the neuroma that forms) has not been examined. Furthermore, little is known regarding the distribution of other members of the shaker type Kv1 channels family such as Kv1.4 and 1.6 following nerve injury.

Here we show that within a neuroma, expression levels of Kv1.1 and 1.2 are markedly reduced but over time Kv1.4 and 1.6 expression increases within juxtaparanodes and paranodes. At sites remote from injury, there is also a gradual redistribution of Kv1 channels to the paranode. Electrophysiological and behavioural experiments suggest that changes in subunit expression and redistribution of Kv1 channels act a 'brake' on the hyperexcitable state that arises in myelinated axons following traumatic nerve injury.

## Results

### Expression of Kv1 channels subunits switches at nodal regions after nerve injury

To investigate the role of Kv1 channels in hypersensitivity after nerve injury we used a model of complete sciatic nerve transection followed by positioning of the proximal stump superficially under the skin of the leg [modified version of *Dorsi et al. (2008)*]. This model enables us to study both the expression of Kv1 channels within the neuroma and undertake behavioural analysis using specific blockers of Kv1 channels.

To study how the localisation of Kv1 channels changes within the nodal complex, we used a pan-Nav channel antibody to label the node of Ranvier, a Caspr antibody to label the paranode and Kv1.2 and Caspr2 antibodies to label the juxtaparanode. In the naïve axon, we observed that over 90% (mean 91 ± SEM 2.9%, n = 4 animals) of the nodes presented a characteristic morphology with Nav clustering in the centre, surrounded by caspr at both sides and Kv1.2 clustered within the juxtaparanode. Of other members of the Kv1 channels family, Kv1.4 and 1.6 are expressed by DRG cells (*Everill et al., 1998*; *Thakur et al., 2014*; *Chiu et al., 2014*). We found that in the naïve state Kv1.4 was only expressed within a very small proportion of nodes (5.5 ± 4.6%, n = 4 animals) within the juxtaparanode and Kv1.6 was not present within the nodal complex (*Figure 1*).

At the site of the neuroma (day 7 and 21), only half of the nodes demonstrated this typical morphology (day 7 = 47.5 ± 5.1%, day 21 = 46 ± 6.%, n = 4 animals per group, 36–64 nodes per animal);

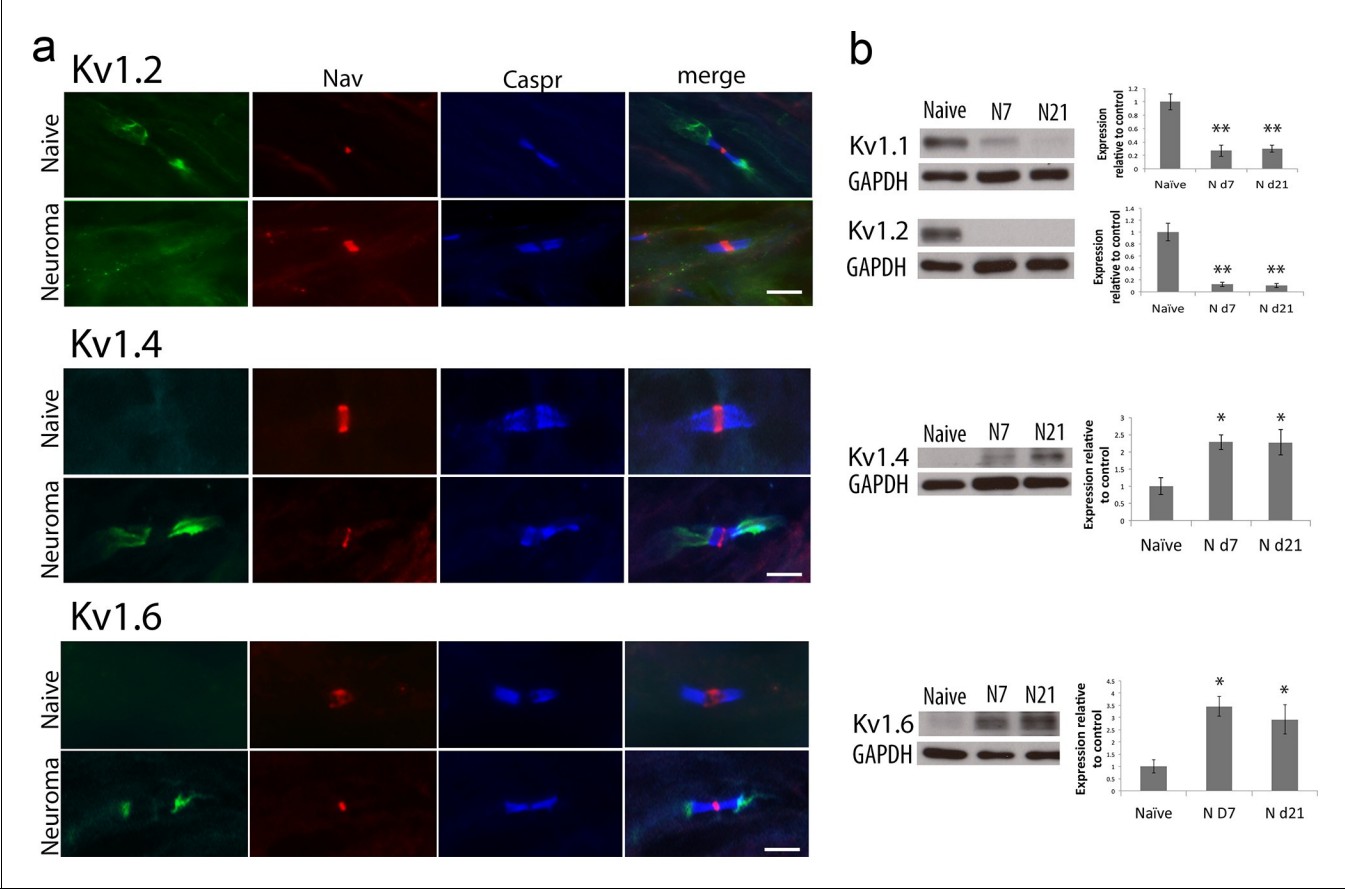

**Figure 1.** Kv1 channels expression in the naïve nerve and 21 days after sciatic nerve axotomy (note that the samples were collected from the neuroma site). (**a**) Representative images of longitudinal nerve sections immunostained with Kv1 channels in green (Kv1.2, Kv1.4 and Kv1.6 respectively), a panNav antibody in red (to identify the node), and caspr in blue (to identify the paranode). Kv1.2 is expressed in the juxtaparanode in naïve nerves but it is not present at 21 days after nerve injury. Kv1.4 and kv1.6 are not present in uninjured nerve but are expressed after nerve injury. Note that when Kv1.4 and Kv1.6 are expressed, they are not confined to the juxtaparanode only but invade the paranode. (**b**) Western blots showing expression of Kv1 channels in the naïve nerve, at 7 and 21 days after axotomy. Kv1.1 and Kv1.2 are expressed in the naïve nerve and down-regulated after axotomy, while Kv1.4 and Kv1.6 have a low/null expression in the naïve nerve and are up-regulated after injury (*p<0.05, **p<0.001, one Way ANOVA, n=6 per group). Scale bars = 5 μm.

The following source data is available for figure 1:

**Source data 1.** Source data for *Figure 1*.

the rest were split (day 7 = 23.3 ± 5.9%, day 21 = 13.5 ± 2.7%), presented as heminodes (caspr at one side only, day 7 = 21.8 ± 4.4%, day 21 = 28.9 ± 5.5%), or were 'naked' (Nav clusters alone, with no caspr, day 7 = 7.2 ± 3%, day 21 = 11.4 ± 3.2%, *Figure 2*). This is in accordance with previous literature examining the localisation of voltage-gated sodium channels (*Henry et al., 2006*; *Thakur et al., 2014*). At day 7 after injury, Kv1.2 channels were not located strictly not only in the juxtaparanodal regions but also overlapped with paranodal proteins. To objectively measure this, we quantified the distance between the Nav channels staining and the distal end of the caspr staining, and the distance between the Nav channels staining and the proximal end of Kv1 channels. The difference between these two distances was indicative of the level of overlap between Kv1 channels and paranodal proteins (note that 'naked' nodes were not included in the analysis of the spatial distribution of Kv1 channels because by definition these only consist of Nav clusters without paranodal and juxtaparanodal proteins). In naïve axons the distance between Nav channels staining and the end of caspr was 3.8 ± 0.2 μm, and the distance between Nav channels staining and the start of Kv1.2 staining was 4.2 ± 0.2 μm, resulting in a relatively small, albeit positive, difference between

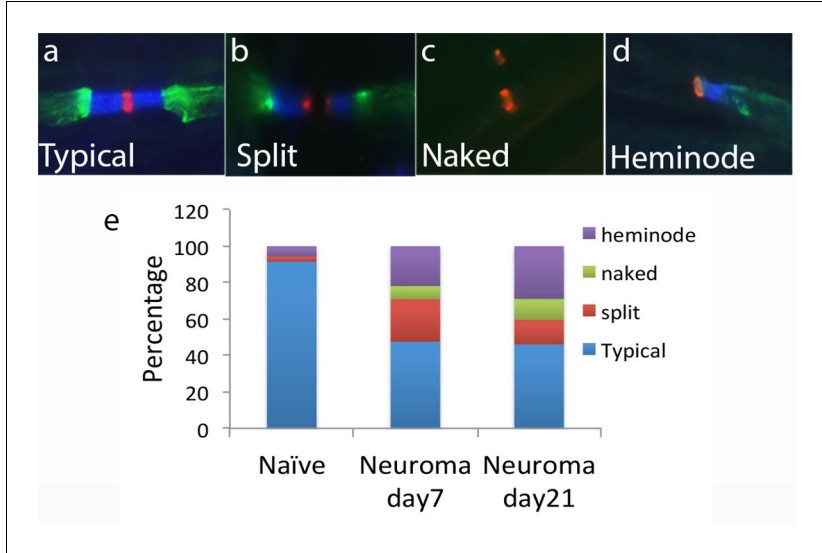

**Figure 2.** Nav channel expression. Representative sections of longitudinal nerves immunostained with Kv1.2 in green, a panNav antibody in red (to identify the node), and caspr in blue (to identify the paranode) from neuroma day 21. (**a**) A typical pattern of Nav expression localized at the node of Ranvier and flanked by caspr staining is shown. The altered forms of Nav channel accumulations seen in the injured nerve included (**b**) split nodes: These were nodes that had two distinct Nav channels accumulations, separated by a gap in the Nav channels staining within the same fibre and with each Nav channels accumulation flanked on one side with caspr staining, or (**c**) naked nodes: those Nav channel accumulations that lacked an association with caspr (**d**) heminodes: nodes where the caspr staining was located on only one side of a contiguous Nav channel accumulation. (**e**) Quantification of different types of sodium cluster accumulation in the naïve state and after nerve injury is shown.

The following source data is available for figure 2:

**Source data 1.** Source data for *Figure 2*.

these distances (0.5 ± 0.08 μm, n = 4 animals, 25–40 nodes per animal), indicating there was no overlap. At day 7 after injury, the distance between Nav channels staining and the end of caspr was 4 ± 0.1 μm, and the distance between Nav channels staining and the start of Kv1.2 staining was 3.2 ± 0.2 μm, giving a negative value for the difference between both distances (-0.8 ± 0.1 μm, n = 3 animals, 32–35 nodes per animal), which indicates that the Kv1 channels were co-localised with caspr staining and moving closer to the node (*Figure 3*). Note that the distance between Nav channels staining and the end of caspr staining remained unchanged after injury, while the distance between Nav channels staining and the start of Kv1.2 staining was significantly reduced. Contactin-associated protein-like 2 (*Caspr2*) forms a complex with Kv1 channels at the juxtaparanode (*Chiu et al., 2014*). We evaluated if caspr2 moves closer to the node together with Kv1 channels. We measured the distance between the Nav channels staining and the distal end of the caspr staining, and the distance between the Nav channels staining and the proximal end of caspr2. In naïve axons, the distance between Nav channels staining and the end of caspr was 3.8 ± 0.3 μm, and the distance between Nav channels staining and the start of caspr2 staining was 4.3 ± 0.0 μm, resulting in a small difference between these distances (0.49 ± 0.09 μm, n = 4 animals, 25–30 nodes per animal), indicating there was no overlap. At day 7 after injury, the distance between Nav channels staining and the end of caspr was 4.1 ± 0.2 μm, and the distance between Nav channels staining and the start of caspr2 staining was 2.8 ± 0.2 μm, giving a negative value for the difference between both distances (-1.2 ± 0.1 μm, n = 3 animals, 30 nodes per animal), which indicates that the caspr2 co-localised with caspr staining and had moved closer to the node together with Kv1 channels (*Figure 3*).

The redistribution of the channels seen could simply be a reflection of direct injury at the site of axotomy. We therefore, studied a site proximal to the neuroma (1 cm) and compared it to the neuroma site. The effect on Kv12 re-localization remains the same at the site far from the neuroma: The difference between Nav channels staining and the end of caspr distance and Nav channels staining

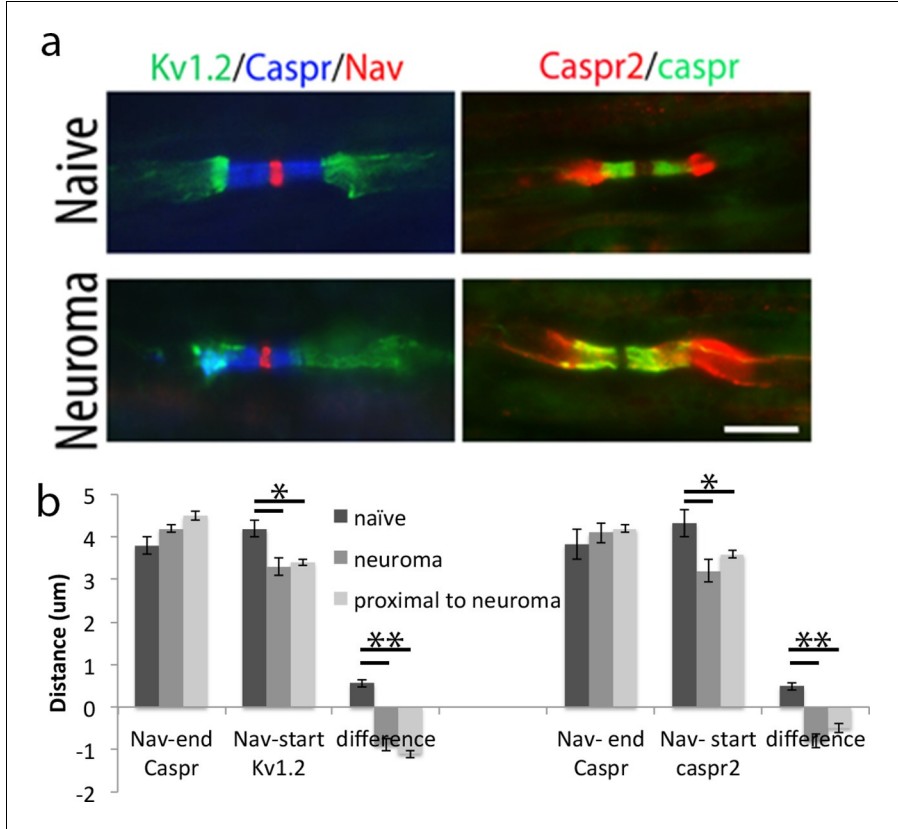

**Figure 3.** Relocalization ok Kv1.2 and caspr2 at 7 days after neuroma. (a) Representative longitudinal sections of nerves immuno-stained with Kv1.2 in green, a panNav antibody in red (to identify the node) and caspr in blue (to identify the paranode). Kv1.2 is expressed in the juxtaparanode in naïve nerves but it also co-localized with caspr staining at 7 days after injury. Representative longitudinal sections of nerves immuno-stained with caspr2 in green and caspr in red. Caspr2 is confined to the juxtaparanode in naïve nerve but co-localized with caspr at 7 days after injury. (b) We quantified the distance between the sodium channel staining (Nav) and the end of the caspr staining, distance between the sodium channel staining (Nav) and the start of the Kv1.2/1caspr2 staining, and difference between these distances. A negative value represents an overlap of paranodal and juxtaparanodal proteins. Note that he distance between the sodium channel staining (Nav) and the end of the caspr staining remains unchanged after nerve injury, while the distance between the sodium channel staining (Nav) and the start of the Kv1.2/caspr2 staining is significantly shortened after nerve injury, indicating co-localization of Kv1.2 and caspr2 with caspr (n = 5 animals, 20–41 nodes per animal), p<0.001, one way ANOVA Tukey post hoc tests). We analyzed uninjured (naïve) nerve, nerve at the site of the neuroma (day 7), and nerve 1 cm proximal to the neuroma (day 7). The effect on Kv1.2 re-localization remains the same at the site far from the neuroma. The effect on caspr2 re-localization is slightly smaller at the site 1 cm proximal to the neuroma compared with the neuroma site, but it is still significantly different from the naïve **p<0.001, *p<0.05, PRN = paranode, JXP = juxtaparanode. Scale bars = 5 μm.

The following source data is available for figure 3:

**Source data 1.** Source data for *Figure 3*.

and the start of Kv1.2 distance was -1 ± 0.09 μm, at the site close to the neuroma and -1.1 ± 0.09 μm, at the site far from the neuroma (p = 0.6, n = 5 animals, 24–26 nodes per animal). The effect on caspr2 re-localization is slightly smaller at the site 1cm proximal to the neuroma compared with the site close to the neuroma, but it is still significantly different from the naïve (p<0.001): The difference between Nav channels staining and the end of caspr distance and Nav channels staining and the start of caspr2 distance was -1 ± 0.1 μm, at the site close to the neuroma and -0.5 ± 0.1 μm, at the site far from the neuroma (p = 0.06, n = 5 animals, 21–26 nodes per animal, *Figure 3c*).

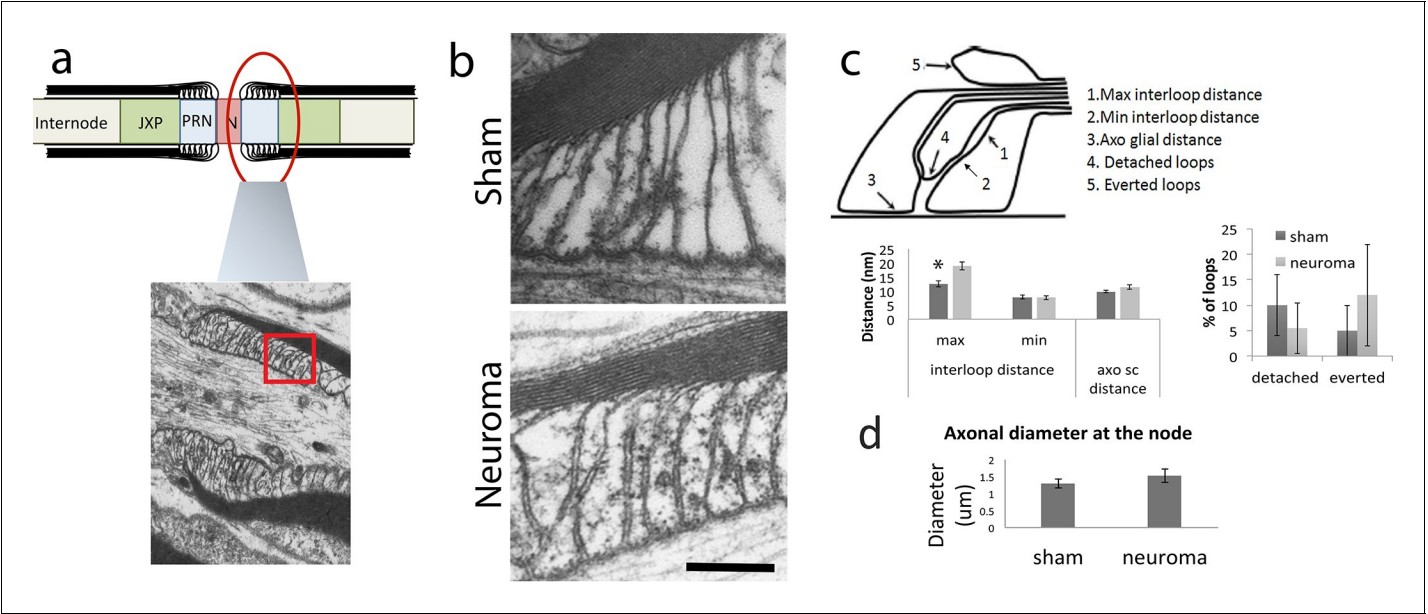

**Figure 4.** Ultrastructural anatomy of the node of Ranvier within the sciatic nerve following axotomy. We used electron microscopy to look at the ultrastructure anatomy of the node. (a) Shows a diagram of the node, paranode and juxtaparanode and a low magnification section of this area in a sham-operated nerve. The red box denotes the area that was used for quantification as seen in **b**. (b) High magnification views of the paranodal loops are shown in the sham and 21 days following axotomy (magnification 135 000x) (c) We quantified different aspects of the attachment of the Schwann cell paranodal loops to the axon. This is illustrated in right panel which denotes the different parameters measured: The maximal and minimal distance between interloops, the distance between the glia and the axon, the number of detached loops and the number of everted loops. We found a significant increase in the maximal distance between loops in the neuroma compared to sham nerves (one way ANOVA, p = 0.005). There were no significant differences in any of the other measurements. (d) We quantified the diameter of the axons at the site of the node and found no difference between the uninjured and injured axons. Scare bars: 200 nm.

The following source data is available for figure 4:

**Source data 1.** Source data for *Figure 4*.

This change on juxtaparanode proteins localisation could be due to a disorganisation of the paranodal axo-glial junctions (paranode loops). Therefore, we examined their ultrastructural anatomy using electron-microscopy and measured the distance between the axon and the paranodal loops (axo-glial distance), the number of detached and everted loops and the minimal and maximal distance between loops. We analysed sciatic nerves from sham-operated and neuroma animals and we observed very few detached or everted loops, with no differences between groups. The close apposition between the axon and the paranodal loop was unchanged as the axo-glial distance was not significantly changed (sham = 9.7 ± 0.4 nm, neuroma = 11.4 ± 0.7 nm, n = 4–5 animals per group, 6–9 nodes per animal one way ANOVA p = 0.08). We observed a small but significant increase in the maximal distance between loops (sham = 12.5 ± 1 nm, neuroma = 18.9 ± 1.4 nm, one way ANOVA, p = 0.005). In summary, these results suggest that although there was no major disruption of the septate axoglial junctions there was a small but significant increased separation between the paranodal loops (*Figure 4*). Note that the axonal diameter at the node did not change (*Figure 4d*).

βII spectrin is a cytoskeletal protein that has recently been shown to be essential for the localization of Kv1 channels to the juxtaparanode and is proposed to form a sub-membranous barrier to lateral diffusion of Kv1 channels into the paranode (*Zhang et al., 2013*). Using IHC, we looked at βII spectrin in the neuroma and found that it is expressed at the paranode and juxtaparanode. We quantified the staining at the paranodal domain and found that the expression of this protein was reduced by more than half compared to naïve (immunofluorescence normalised to naïve: 0.4 ± 0.02, p<0.001, t-test, n = 50–83 heminodes, a–b). βII spectrin is also expressed in the sub-membranous regions of Schwann cells where it does not appear to change with nerve injury. To further quantify the change of expression of this protein in the neuron, we performed Western blotting in the soma

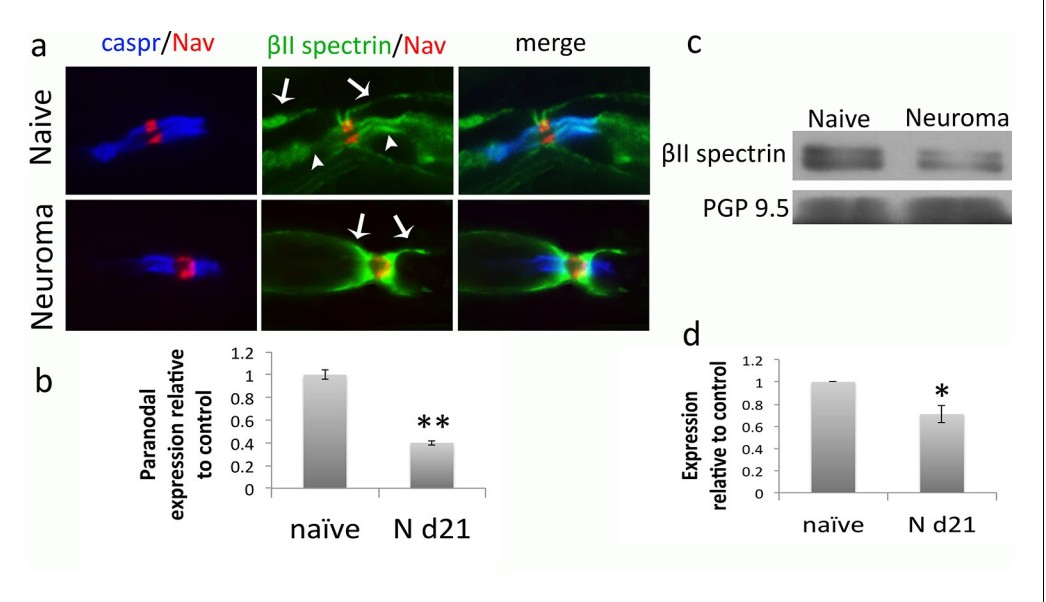

**Figure 5.** βII spectrin expression in naïve and neuroma nerves. (a) Representative sections of longitudinal nerves immunostained with βII spectrin in green, a panNav antibody in red (to identify the node), and caspr in blue (to identify the paranode). βII spectrin is expressed both in the surface of Schwann cells (arrows) and in the axon at the paranodal and juxtaparanodal region (arrow heads) in naïve nerves. At 21 days after axotomy (neuroma), βII spectrin can be only seen in the Schwann cell (arrows) but not in the axonal domains. (b) Quantification of βII spectrin immunofluorescence in the paranode (identified by caspr staining) showing a significant reduction in neuroma versus naïve (immunofluorescence normalised to naïve: 0.4 ± 0.02, p<0.001, t-test, n = 50–83 heminodes). (c) Western blots showing expression of βII spectrin in the DRG of naïve and day 21 neuroma. (d) Quantification of WBs. Expression of βII spectrin in the DRG was reduced by 30% after nerve injury. PGP9.5 was used as a loading control (expression relative to naïve: 0.7 ± 0.08, p = 0.04, t-test). **p<0.001, *p<0.05. Error bars denote SEM.

The following source data is available for figure 5:

**Source data 1.** Source data for *Figure 5*.

---

of sensory neurons (dorsal root ganglia- DRG) and observed a 30% decrease following nerve injury compared to naive (expression relative to naïve: 0.7 ± 0.08, p = 0.04, t-test, n = 4, *Figure 5c–d*).

At day 21 after injury, in marked contrast with the naïve axons, very few of the nodes at the neuroma site (day 21) showed Kv1.2 immunostaining (8.3 ± 0.8% in neuroma vs. 86.1 ± 4.4% in naive, n = 4 animals per group, 25 nodes per animal p<0.001 t-test). Conversely, most of the nodes at the neuroma site (day 21) showed intense Kv1.4 immunostaining (73.3 ± 12% in injured vs. 5.5 ± 4 in naive, n = 4 animals per group, 30 nodes per animal p<0.001 t-test) and Kv1.6 immunostaining (66.6 ± 14% vs. in injured vs. none in naive, n= 4 animals per group, 30 nodes per animal p<0.001 t-test) (*Figure 1*). (Note that naïve nerves were comparable in terms of quantification to nerves from sham-operated animals [p = 0.48]).

We used Western blotting to quantify the expression of the different α subunits and found that Kv1.1 and Kv1.2 expression were significantly reduced at days 7 and 21 following nerve injury, while Kv1.4 and Kv1.6 were significantly upregulated at the neuroma site (*Figure 1*). We looked at Kv1 channel expression in the DRG using IHC and we observed that Kv1.2 expression is reduced after injury, while Kv1.4 and Kv1.6 expression remains unchanged (*Figure 6a*) (this is at the DRG soma, although expression could be seen to increase within paranodes/juxtaparanodes after injury). We also quantified protein expression within the DRG over the same time course using Western blot analysis. The expression of Kv1.2 was significantly reduced following nerve injury (*Figure 6b–e*) consistent with previous findings, (*Everill et al., 1998*; *Ishikawa et al., 1999*; *Kim et al., 2002*; *Yang et al., 2004*), and there was a trend for a reduction in Kv1.1 although this did not reach significance. The expression of Kv1.4 and 1.6 within the DRG did not significantly change following injury

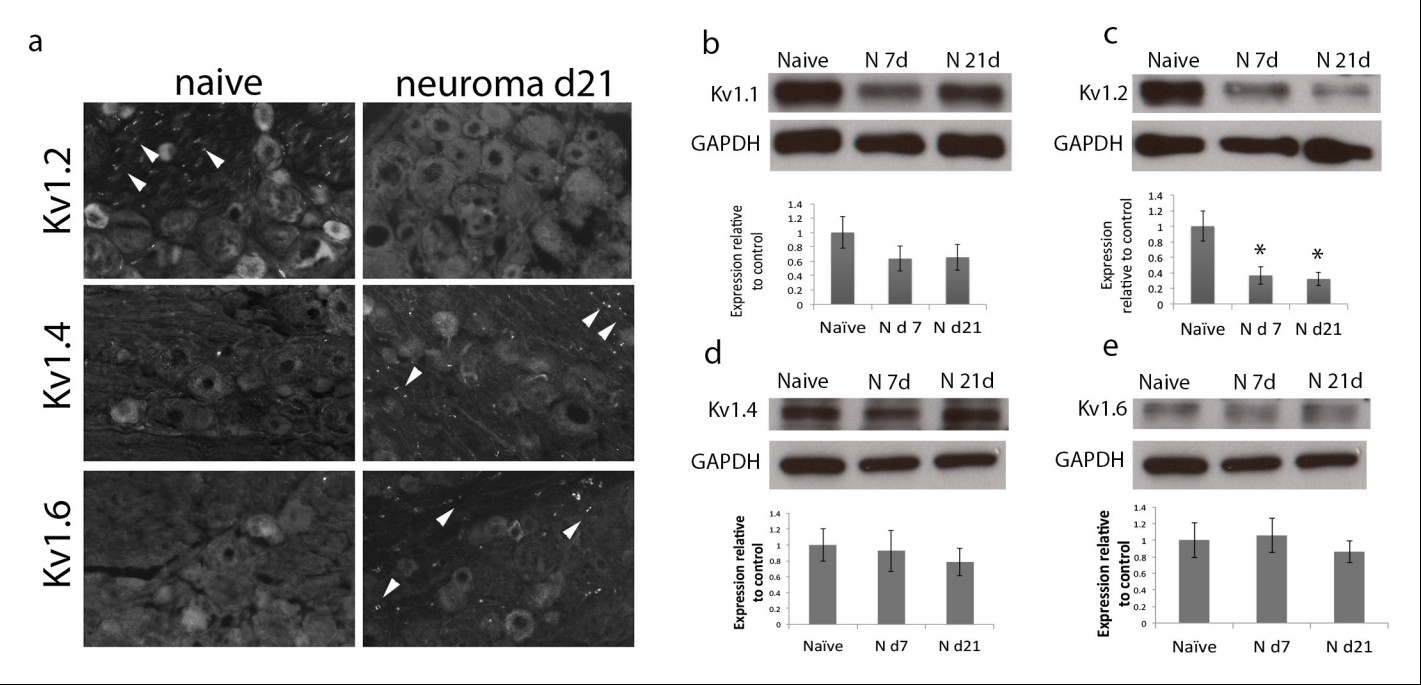

**Figure 6.** Expression of Kv1 channels in the DRG in the naïve state, 7 and 21 days after axotomy (neuroma). (**a**) Representative sections of naive and neuroma day 21 DRG immunostained with Kv1.2, Kv1.4, and Kv1.6. Note that Kv1.2 expression in DRG cells and axonal juxtaparanodes (arrow heads) is reduced after injury, while Kv1.4 and Kv1.6 expression remains unchanged in DRG cells, and it is present in axonal juxtaparanodes (arrow heads) after injury. In each panel (**b–e**), a representative blot is shown for each time point with GAPDH as a loading control. Quantification of 6 animals per condition is shown below (**b,d,e**) Expression of Kv1.1, Kv1.4 and Kv1.6 within the DRG does not significantly change after axotomy. (**c**) Kv1.2 expression is significantly decreased after axotomy. (*p<0.05, one Way ANOVA, n = 6 per group).

The following source data is available for figure 6:

**Source data 1.** Source data for *Figure 6*.

(one Way ANOVA, n = 6 per group) suggesting that increased expression within the juxtaparanode and paranode at the neuroma site is a likely consequence of altered trafficking of these proteins rather than global changes in expression.

We next looked into human nerve tissue to see if these changes were relevant to patients with neuropathic pain We collected 6 control samples (from subjects having their sural nerves removed to use as a bridge for hand reconstructive surgery) and 6 samples obtained from patients undergoing removal of Morton's neuroma (interdigital nerve entrapment neuropathy). IHC (n = 3 per group, 8–10 nodes per patient) showed that only Kv1.2 is expressed in the juxtaparanode of healthy subjects (90 ± 10% of nodes were Kv1.2 positive) with absent Kv1.4 and 1.6 staining as observed in the rat. However, in neuroma Kv1.2 expression in the juxtaparanode was minimal (13.3 ± 8.1%) whilst Kv1.4 and Kv1.6 were expressed in most of the nodes (92.5 ± 7.4% for Kv1.4; 73.5 ± 8.8 for Kv1.6) (*Figure 7a–b*). We used western blotting to quantify Kv1 channels proteins in the nerves of the patients (n = 6 per group) and found that Kv1.2 expression was significantly decreased in neuroma compared to control nerve (to 0.48 ± 0.1 of the control, Mann-Whitney U-Test, p 0.005). In contrast, expression of Kv1.4 and Kv1.6 were significantly increased in neuroma compared to controls (to 6.3 ± 3.5, and 9.4 ± 6.6 of the control respectively, Mann-Whitney U-Test p = 0.005 both) (*Figure 7c–d*).

## Kv1 channels change their distribution in the nodal regions at sites distant from the injury

We investigated the localisation of Kv1 channels at a site distant from the injury site. To do so, we used a model of L5 spinal nerve transection (SNT) (*Kim and Chung, 1992*) and studied the dorsal

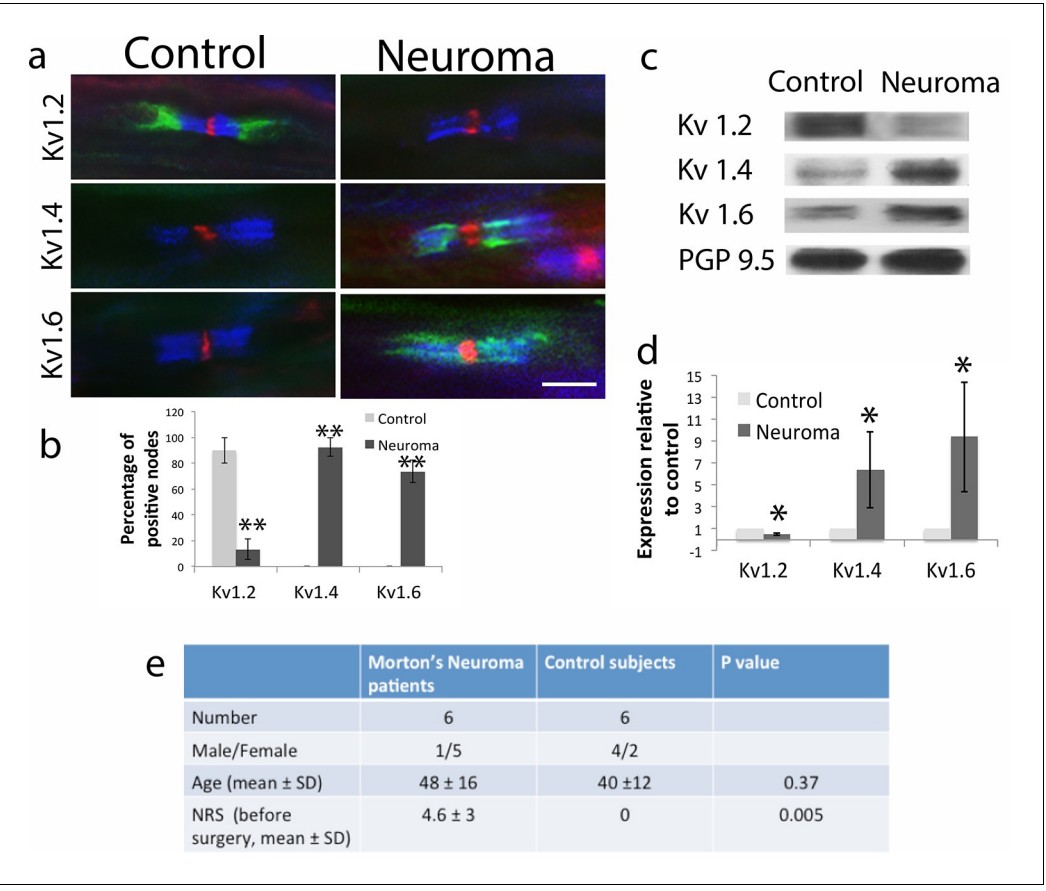

**Figure 7.** Kv1 channels expression in the sural nerve of healthy volunteers (control) and from patients with painful Morton neuroma. (a) Representative sections of longitudinal nerves immunostained with Kv1 channels in green (Kv1.2, Kv1.4 and Kv1.6 respectively), a panNav antibody in red (to identify the node), and caspr in blue (to identify the paranode). Kv1.2 is expressed in the juxtaparanode in control nerves but it is not present in the injured nerve. Kv1.4 and kv1.6 are not present in control nerve but are expressed in neuroma within the juxtaparanode and encroaching on the paranode nodes. (b) Quantification of the percentage of Kv1.2, Kv1.4, and Kv1.6 positive nodes in control and neuroma nodes (n = 3 per group, one way ANOVA). (c) Western blots showing expression of Kv1 channels in control and neuroma nerve. (d) Quantification of WBs (n = 6 per group, one way ANOVA). Kv1.2 is expressed in the control nerve and down regulated after axotomy, while Kv1.4 and Kv1.6 have a low/null expression in the control nerve and are up-regulated in neuroma. PGP9.5 was used as a loading control. Error bars denote (e) Patients and control subjects demographic data. The female/male ratio is different in patients with Morton neuroma and controls (patients with traumatic lesion of the hand) reflecting the F/M ratio of these different conditions. The mean age of patients with Morton neuroma is slightly higher than in control subjects, although it is not significant (t-test). All patients with Morton neuroma presented with pain (mean NRS 4.6), while control presented no pain in the area innervated by the sural nerve (Mann Whitney test). NRS: numerical rate score. Error bars denote SEM. Scale bars = 5 μm, **p<0.001, *p<0.05.

The following source data is available for figure 7:

**Source data 1.** Source data for *Figure 7*.

roots (ie. proximal to the DRG). We used this model instead of the neuroma model to have certainty that all the dorsal root axons studied had their peripheral terminals injured.

In the dorsal roots from naïve animals, the distance between Nav channels staining and the end of caspr was 3.5 ± 0.2 μm, and the distance between Nav channels staining and the start of Kv1.2 staining was 4 ± 0.2 μm, resulting in a small positive difference between these distances (0.5 ± 0.1 μm, n = 4 animals, 25–32 nodes per animal), indicating there was no overlap. Seven days after transection of L5 spinal nerve this distance was still positive (0.24 ± 0.05 μm, n = 4 animals; distance

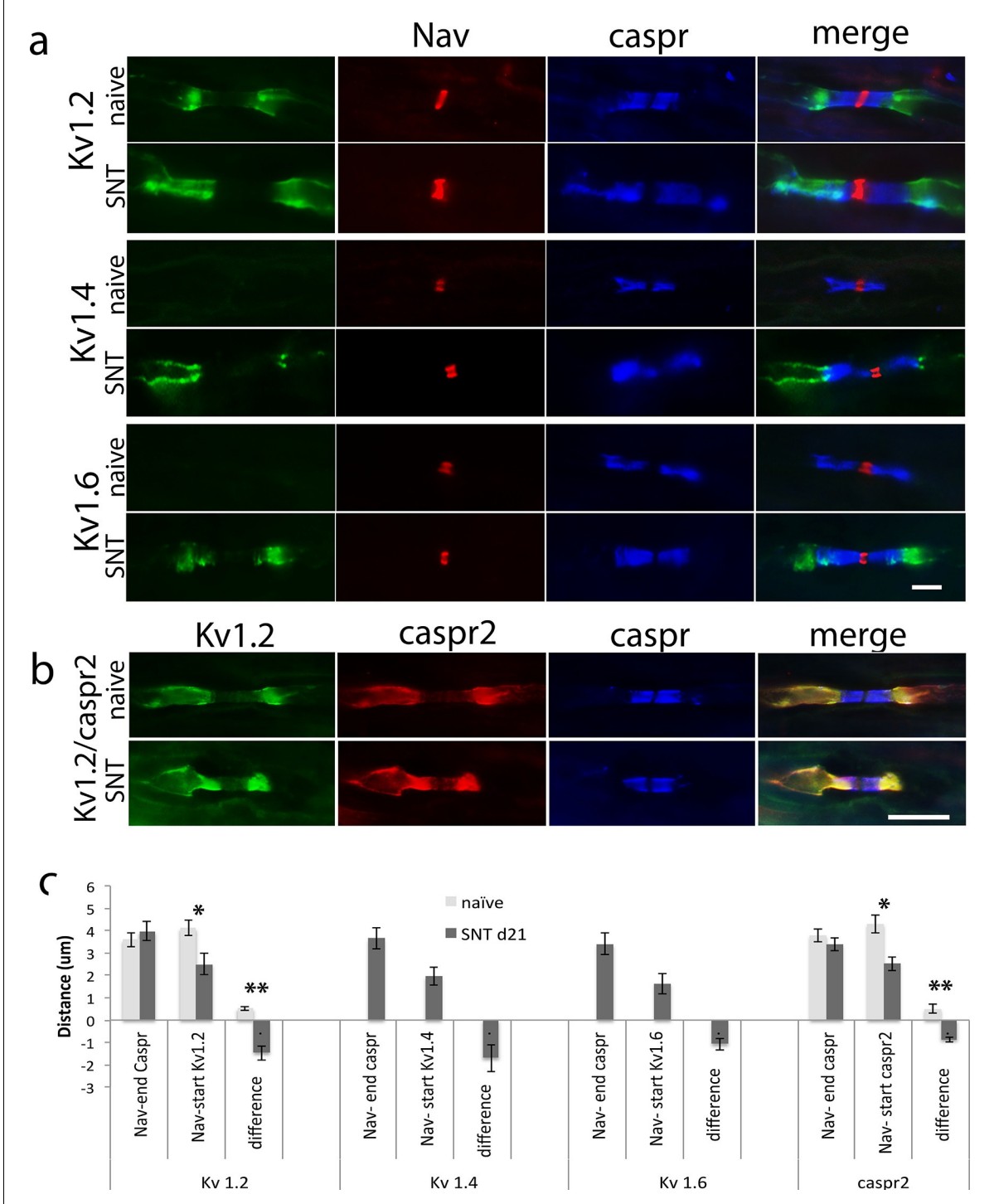

**Figure 8.** Kv1 channels expression in the dorsal roots of naïve animals and 21 days after spinal nerve transection (SNT). (**a**) Representative sections of longitudinal dorsal roots immunostained with Kv1 channels in green (Kv1.2, Kv1.4 and Kv1.6, respectively), a panNav antibody in red (to identify the node), and caspr in blue (to identify the paranode). Kv1.2 is expressed only in the juxtaparanode in naïve nerves but after injury it invades the paranode. Kv1.4 and kv1.6 are not present in uninjured nerve but are expressed within the juxtaparanode after nerve injury and also invade the paranode. (**b**) Re-localization of caspr2 at 21 days after spinal nerve transection (SNT). Representative sections of longitudinal L5 dorsal roots immuno-stained with caspr2 in red, Kv1.2 in green, and caspr in blue. Kv1.2 and caspr2 are expressed in the juxtaparanode in naïve nerves but co-localized with caspr staining at 21 days after injury. (**c**) Quantification of: distance between the sodium channel staining (Nav) and the end of the caspr staining, distance between the sodium channel staining (Nav) and the start of the Kv1.2/1.4/1.6/caspr2 staining, and difference between these distances. A negative value in this difference represents an overlap of paranodal and juxtaparanodal proteins. Note that the distance between the sodium channel staining (Nav) and the

*Figure 8 continued on next page*

Figure 8 continued
end of the caspr staining remains unchanged after nerve injury, while the distance between the sodium channel staining (Nav) and the start of the
Kv1.2/1.4/1.6/caspr2 staining is significantly shortened after nerve injury. Kv1.4 and Kv1.6 were absent in naïve (n = 5 animals/4 sections per animal,
*p<0.05, **p<0.001). Scale bars = 5 µm.
The following source data is available for figure 8:

**Source data 1.** Source data for *Figure 8*.

Nav-end caspr 3.7 ± 0.2 µm, distance Nav-start Kv1.2 3.9 ± 0.1 µm; 30–40 nodes per animal). However, at 21 days after nerve injury, we noted a significant overlap between the end of caspr staining and the start of the Kv1.2 staining (-1.4 ± 0.3 µm, n = 4 animals, 38–40 nodes per animal, one way ANOVA, p<0.001; distance between Nav-end caspr 3.9 ± 0.4 µm, distance between Nav-start Kv1.2 2.5 ± 0.4 µm). We observed this novel localisation of Kv1.2 in the paranode which (in contrast to the neuroma site) is still clearly present after nerve injury within the dorsal root. We also observed novel expression of Kv1.4 and 1.6 in the dorsal root of injured animals, and these were localised to the paranode in addition to the juxtaparanode (the distance between end of caspr and start of Kv1.4 and 1.6 staining was -1.7 ± 0.6 µm and -1.07 ± 0.2 µm respectively; for Kv1.4 distance between Nav-end caspr 3.6 ± 0.4 µm, distance between Nav-start Kv1.4 1.9 ± 0.6 µm; for Kv1.6: distance between Nav-end caspr 3.4 ± 0.4 µm, distance between Nav-start Kv1.6 1.6 ± 0.4 µm n = 4 animals per group, 30–35 nodes per animal *Figure 8*).

Contactin-associated protein-like 2 (*Caspr2*) is normally localized at the juxtaparanode and associates with K+ channels (*Chiu et al., 2014*). Interestingly, we observed that caspr2 is mobilized into the paranode regions in a similar way to Kv1 channels; the difference between the Nav-caspr staining distance and the Nav-caspr2 distance is minimal in naïve roots (0.5 ± 0.1 µm; distance between Nav-end caspr 3.8 ± 0.3 µm, distance between Nav-start caspr2 4.3 ± 0.4 µm, 30–38 nodes per animal 4 animals), and after SNT, this distance becomes negative (-0.88 ± 0.1 µm, n = 5 animals; distance between Nav-end caspr 3.4 ± 0.3 µm, distance between Nav-start caspr2 2.5 ± 0.3 µm 35–40 nodes per animal p<0.001, t-test) indicating an overlap between caspr and caspr2 immuno-labeling (*Figure 8*).

## Effect of Kv1 channels redistribution and change in expression on the incidence of spontaneous activity

Spontaneous activity in naïve axons was present in less than 5% of A-fibres (4.5%). We assessed the effect of blocking the Kv1 channels using α-DTX, a toxin isolated from black and green mamba snakes which is a selective and effective blocker of Kv1-containing oligomers composed of Kv1.1, Kv1.2, or Kv1.6 subunits (*Harvey, 2001*). We applied the toxin at the neuroma site (or, in control animals, acutely cut sciatic nerve stump) and to the L5 DRG and recorded from sensory axons in thin strands dissected from the dorsal root. In the naïve situation (n = 222 neurons, 10 animals, *Figure 9a*), the incidence of spontaneous activity did not significantly change after α-DTX (5.4 and 9.8% of myelinated afferents when the toxin was applied to nerve stump or L5 DRG, respectively). Two days after transecting the sciatic nerve (n = 291 neurons, 4 animals), spontaneous activity at the injured nerve increased to 22% of myelinated afferents, and it was similar with or without toxin (toxin applied to neuroma 26.1%, toxin applied to the L5 DRG 26%). However, at 7 days after nerve injury (n = 241 neurons, 7 animals) the proportion of spontaneously active afferents had decreased to 6.2%, and application of the toxin now induced a significant increase in proportion of spontaneously firing myelinated afferents (11.2% toxin at the neuroma p = 0.07, and 15.6% toxin to L5 DRG, p = 0.002, chi-square test). At day 21 after injury (n = 237 neurons, 7 animals) spontaneous activity has decreased to baseline levels (2.5% of afferents within the dorsal root), but application of the toxin to both neuroma site or L5 DRG significantly increased the proportion of myelinated afferents demonstrating spontaneous activity (7.2% p = 0.03 and 17.7 ±% p<0.001 respectively, chi-square test) (*Figure 9a*). In summary, we observed an acute increase in spontaneous activity following nerve injury, which was reversed with time. However, at this later time, blockade of Kv1 channels (which had no effect in the naïve state) could reinstate spontaneous activity almost to levels seen acutely

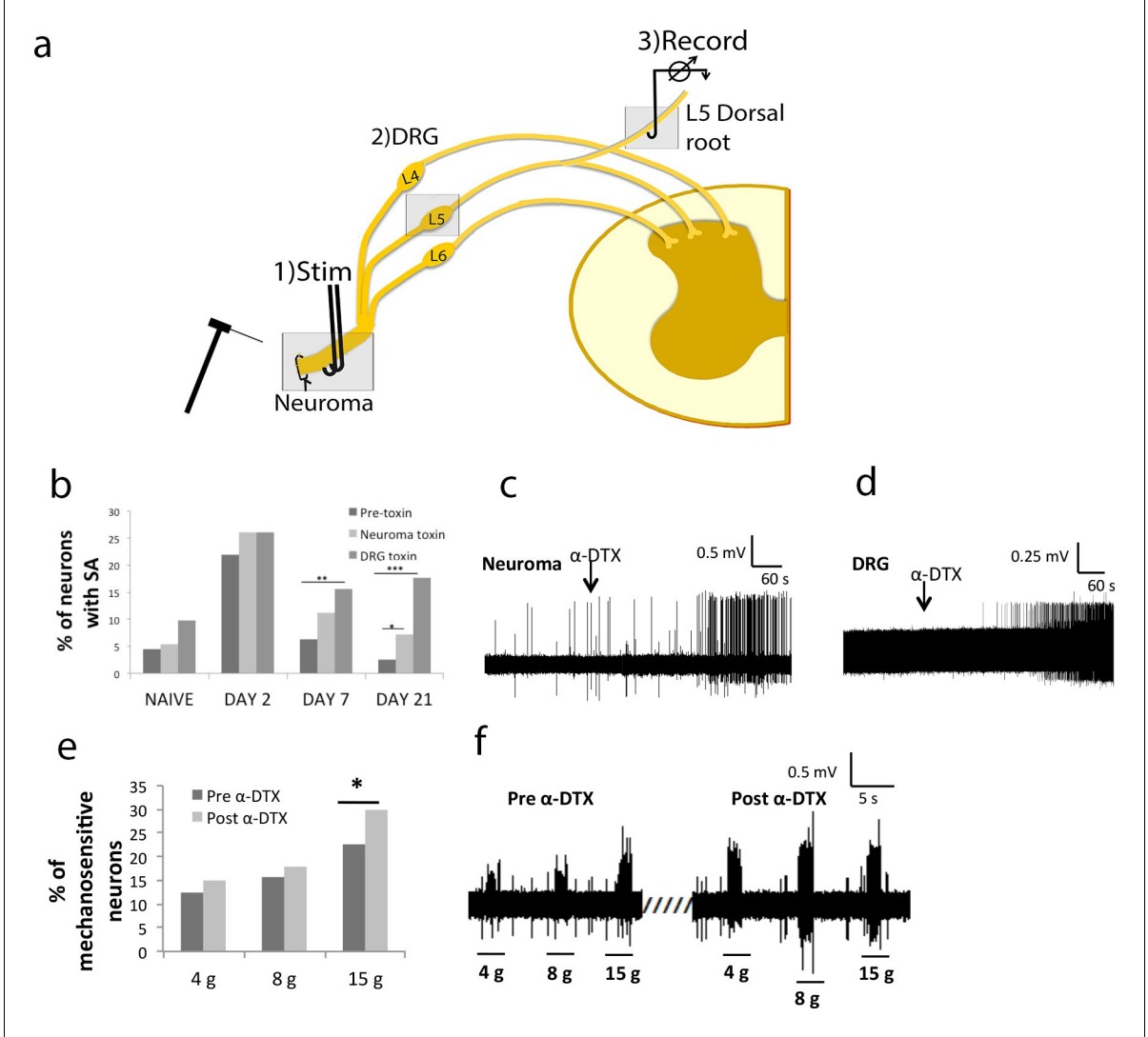

**Figure 9.** Local application of α-DTX reinstates primary afferent hyperexcitability at later time points following nerve injury. (a) Schematic illustration of 3-chamber recording system. 1) Recording chamber, 2) middle chamber, 3) stimulating chamber. The toxin was applied in chambers 1 or 2, respectively. (b) Following sciatic nerve transection, there is a large increase in the proportion of primary afferents demonstrating spontaneous activity at day 2, which is suppressed at days 7 and 21 post injury; Local application of α-DTX to the neuroma and L5 DRG at these later time points (days 7 and 21) significantly increases the proportion of afferents, which are spontaneously active (total proportions per group, chi-square tests, *p<0.05, **p<0.01, ***p<0.001) (c) neuroma application and (d) DRG application of α-DTX. Both recordings were carried out 21 days post-surgery. (e) In the presence of α-DTX, significantly more neurons respond to mechanical stimulation at the neuroma site using a 15 g von Frey filament (*p<0.05; total proportions per group, chi-square tests, all neuroma day 21). (f) Representative traces showing greater responsiveness to mechanical stimulation with von Frey filaments after local α-DTX application.

The following source data is available for figure 9:

**Source data 1.** Source data for *Figure 9*.

after nerve injury. Therefore, Kv1 channels appear to have a role in the recovery from increased excitability following nerve injury (*Figure 9b–d*).

## Effect of Kv1 channel redistribution and change in expression on mechanical sensitivity

Because Kv1 channels have been shown to have a crucial role in mechano-sensitivity (*Hao et al., 2013*), we tested their role in hypersensitivity following nerve injury. In the presence of α-DTX,

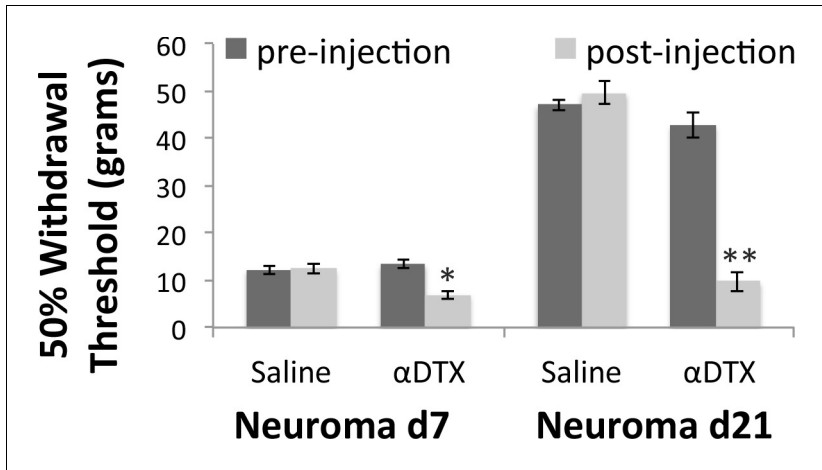

**Figure 10.** Mechanical hypersensitivity is restored by blocking Kv1 channels. Mechanical withdrawal thresholds were assessed by applying a range of Von Frey hairs to the skin over the neuroma site (labelled with a suture). Animals were randomised to receive either subcutaneous αDTX or saline 30 min before testing. Hypersensitivity after nerve injury is very pronounced until day 7, when it slowly starts recovering. At this time point, perineuromal application of αDTX reversed this early recovery. At 3 weeks after nerve injury hypersensitivity is much recovered and perineuromal injection of αDTX restored mechanical hypersensitivity to levels seen acutely after injury (RM two way ANOVA, *p<0.05, **p<0.001).

The following source data is available for figure 10:

**Source data 1.** Source data for *Figure 10*.

significantly more neurons responded to mechanical stimulation using a 15g von Frey filament at the neuroma site at day 21 (with 4 g: pre 17.9% post 21.8%; with 8 g: pre 22.1% post 26%; with 15 g: pre 24.7%, post 30.5%, p = 0.006, chi-square test; n = 259 neurons; *Figure 9e–f*).

## Kv1 channels are responsible for the partial recovery of mechanical hypersensitivity seen chronically after nerve injury

We subsequently used behavioural measures to examine mechanical sensitivity after nerve injury. For this purpose, we used the sciatic neuroma model used before in which the sciatic nerve of rats is transected and the proximal end was sutured superficially below the skin on the animal's leg. We applied von Frey filaments of increasing forces to the site of the skin covering the neuroma. To test the role of Kv1 channels on hypersensitivity, we applied α DTX subcutaneously at the site of neuroma. At baseline, the withdrawal threshold was high and did not change with the application of α DTX (vehicle: 142.9 ± 13 g, toxin 150 ± 17.7 g, n = 9 animals per group). Three days after injury, this threshold dropped to 8.2 ± 0.6 g and was not changed by applying the toxin (9.7 ± 1.1 g) (n = 9 animals per group). However, with time this threshold began to normalise reaching 12.9 ± 0.6 g at 7 days (n = 8) and 44.8 ± 1.6 g at 21 days (n = 8). When we injected αDTX, this recovery was not seen and thresholds stayed low (day 7: 6.9 ± 0.9 g, p = 0.003, n = 9; day 21: 13 ± 2.1 g, p<0.001, n = 7, RM two way ANOVA *Figure 10*).

The time point when the mechanical hypersensitivity began to recover in injured animals coincides with the time when Kv1.1 and Kv1.2 are reduced but Kv1.4 and Kv1.6 are being expressed. α-DTX is a selective blocker for Kv1.1, Kv1.2 and Kv1.6 but has little activity against Kv1.4. CP 339818 hydrochloride however, which is a selective blocker of Kv1.3 and Kv1.4 (*Nguyen et al., 1996*), did not have any effect on mechanical hypersensitivity at early or late time-points post injury suggesting that Kv1.4 is dispensable for the suppression of hyperexcitability. (baselines: saline 133 ± 21 g, CP339818 164 ± 38 g; neuroma day 3: saline 51 ± 2 g, CP339818 44 ± 5 g; neuroma day 7: saline 41 ± 3 g, CP339818 39 ± 4 g; neuroma day 21: saline 73 ± 13 g, CP339818 91 ± 4 g, RM two way ANOVA, p>0.05, n = 8–7 for saline at day 21]).

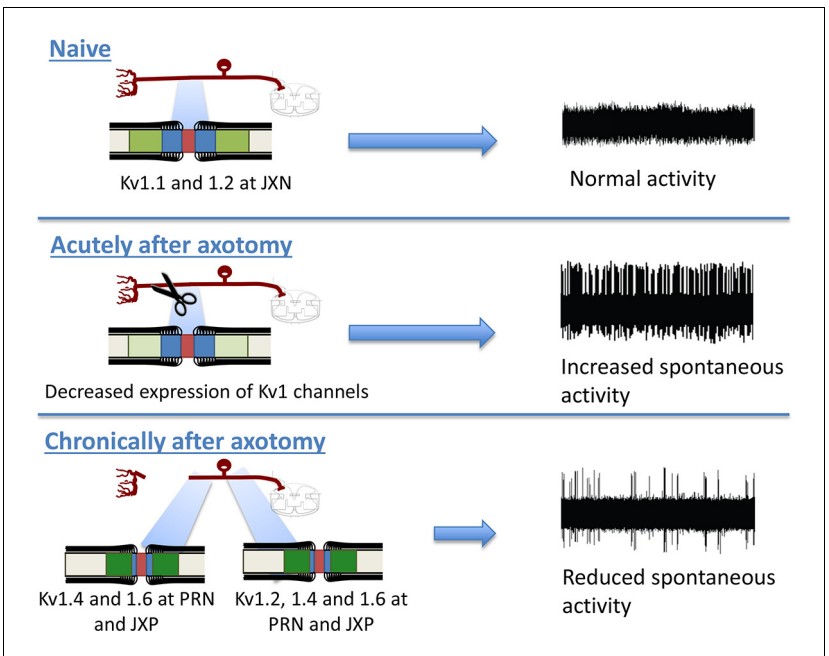

**Figure 11.** Schematic illustration of the changes in Kv1 channels subunit composition and distribution within the nodal complex and the relationship to hyperexcitability. In the naïve state, Kv1 channels (Kv1.1 and 1.2 shown in green) are localised to the juxtaparanode and separated from voltage-gated sodium channels at the node (red) by the paranode (blue). Acutely following axotomy myelinated primary afferents show a marked increase in spontaneous activity as a consequence of complex changes in increased pro-excitatory drive (for instance from voltage-gated sodium channels) as well as reduced 'brakes' on excitability At later stages within the neuroma, although there is less expression of Kv1.1 and Kv1.2, the expression of Kv1.4 and 1.6 increases. Remote from the injury within the dorsal root expression of Kv1.1 and 1.2 is maintained and the expression of Kv1.4 and Kv1.4 also increases. Furthermore Kv1 channels are redistributed to the paranode as well as being expressed within the juxtaparanode. These changes are associated with a suppression of hyperexcitability.

## Discussion

We have focussed on the distribution of Kv1 channels within the axolemma of myelinated axons in rodent models of neuropathic pain as well as in human neuroma tissue. We show that both at the site of injury (using the neuroma model) and within the dorsal root (remote from the injury site) that there is a major change in the α-subunit composition of Kv1 channels (with increased expression of Kv1.4 and 1.6) and furthermore that Kv1 channels are no longer restricted to the juxtaparanode but also re-distribute to the paranode following injury. Blockade of these ion channels using α-DTX reveals that following axonal injury Kv1 channels act to suppress axonal hyperexcitability and hence hypersensitivity to sensory stimuli (*Figure 11*).

We initially examined the distribution of Kv1 channels within the juxtaparanode of axons within the neuroma at the the injury site. Following traumatic nerve injury there is an initial inflammatory response, demyelination and axonal die-back within the proximal nerve stump. This is followed by axonal sprouting (*Fried and Devor, 1988*; *DeFelipe, 1991*) (which is often misdirected) as well as proliferation of connective tissue and glial cells. This can lead to a region of nerve swelling, a neuroma, the development of which is well documented in patients (*Scadding, 1981*; *DeFelipe, 1991*; *Young, 1942*). Nerve injury is often associated with ongoing pain, dysaesthesia and evoked pain; many patients report that touching the skin overlying or compression of the neuroma site can elicit pain and dysaesthesia (*Nikolajsen et al., 2010*). Unfortunately in many patients, the treatment of neuropathic pain remains inadequate (*Finnerup et al., 2015*) and is associated with significant disability (*Stokvis et al., 2010*). Neuroma pain can be modelled in the rodent using a modified version of the tibial transposition neuroma model (*Dorsi et al., 2008*).

## Altered distribution of Kv1 channels in nodal complexes of the neuroma and dorsal root

It has previously been documented that within a neuroma nodes of Ranvier become disorganised (*Levinson et al., 2012*) and we confirm that here with the majority nodes showing abnormalities such as being elongated, split, heminodes or showing Na$_v$ in the absence of caspr. Voltage-gated sodium channels are known to accumulate particularly within axon tips and on denuded axons of the neuroma (*England et al., 1996*). Much less is known regarding the distribution of Kv1 channels changes within the neuroma. This is an important issue as such channels could potentially act as 'brakes' on excitability. At an early time point after injury, we found that Kv1.2 is no longer confined to the juxtaparanode but extended into the paranode. As this could be only a reflection of direct injury, we also looked into a site in the injured nerve 1 cm proximal to the neuroma and found similar changes in distribution, suggesting this is likely to reflect widespread changes within the axon/axoglial signalling (further supported by changes within the dorsal root and discussed below). At a later time-point following injury, we found a striking reduction in the expression of Kv1.1 and 1.2 which are normally localised to the juxtaparanode. In contrast Kv1.4 and 1.6 which are present at a low level in the naïve state are up-regulated and are present both in the paranode and the juxtaparanode. We found broadly similar changes in rodent and human neuroma. This altered expression and localisation is likely to partially reflect the altered relationship between axons and myelinating Schwann cells. Within the neuroma new nodes of Ranvier will be formed as a consequence of myelination of new axon sprouts and remyelination of denuded axons (*Dorsi et al., 2008*; *Dyck et al., 1985*). During developmental myelination and remyelination following primary demyelination (in which the axon remains intact) Kv1.1 and 1.2 can at early time points be observed in other regions apart from the juxtaparanode (at the node of Ranvier and the paranode) before being restricted to the juxtaparanode as the nodal complex matures (*Rasband et al., 1998*; *Vabnick et al., 1999*; *Poliak et al., 2001*). Kv1.4 and 1.6 expression has not to our knowledge been examined during myelination/remyelination.

We also studied the dorsal root, which is remote from the injury site to establish whether there were changes in Kv1 channels composition of the juxtaparanode that reflect the general response of the axon to injury rather than local effects such as inflammation and remyelination at the injury site. We also found striking changes within the nodal complex of sensory axons within the dorsal root. Kv1.2 was no longer down-regulated as had been noted at the neuroma site but its localisation changed following injury: They could be observed in the paranode as well as the juxtaparanode. In the naïve state, very little Kv1.4 and 1.6 expression was noted in the juxtaparanode of the dorsal roots. This is in agreement with previous studies that reported a low frequency of Kv1.4 immunoreactive juxtaparanodes (*Everill et al., 1998*) and no expression of Kv1.6 (*Utsunomiya et al., 2008*). We found however that nerve transection led to markedly increased expression of these α-subunits and they were localised both to juxtaparanode and paranode.

## Factors governing localisation of Kv1 channels to axonal domains in the naïve and injured state

What factors are responsible for the altered distribution of Kv1 channels within the juxtaparanode? We used ultrastructural examination of the juxtaparanode in the dorsal root to examine whether structural changes within the paranode could explain the movement of Kv1 channels into this region (from which they are normally excluded). The transverse bands are important points of attachment between the axon and the paranodal loops of the Schwann cell. These axoglial septate-like junctions are formed by the interaction of caspr (*Bhat et al., 2001*) and contactin (*Boyle et al., 2001*) on the axolemma binding with the 155Kd isoform of Neurofascin expressed on the Schwann cell paranodal loops (*Tait et al., 2000*). These junctions act as diffusion barriers between the nodal and juxtaparanodal membrane. Mice lacking caspr (*Bhat et al., 2001*), contactin (*Boyle et al., 2001*) or NF155 (*Pillai et al., 2009*; *Sherman et al., 2005*) have absent transverse bands, increased distance between the axon and Schwann cell membrane, disorganisation of the paranodal loops and probably as a consequence of the loss of this lateral diffusion barrier Kv1 channels are noted to extend into the paranode. Similarly in mice lacking ceramide galactosyl transferase in which all-putative adhesion components of the paranodal junction are lacking, Kv1.2 and caspr2 are also mis-localised to the paranodes (*Poliak et al., 2001*; *2003*). On ultrastructural examination of the paranodes within the

sciatic nerve following injury, we did not see major structural changes and there was no increase in the distance between the axon and the Schwann cell membrane at the site of attachment of the paranodal loops. We noted an increase in the maximum distance between paranodal loops, which is unlikely to alter the ability of molecules to passively diffuse between the membrane domains of the juxtaparanode and paranode (however it will increase the diameter of the helical pathway between paranodal loops connecting the extracellular space to the axonal internode. One potential consequence of which would be reduced passive resistance to current flow between the node and voltage-gated potassium channel (VGKC) in the juxtaparanode and paranode, which could then have a greater influence on nodal excitability (*Shroff et al., 2011*). A recent publication has demonstrated that in mice lacking βII spectrin expression in axons Kv1 channels were no longer restricted to the juxtaparanode but could also be observed in the paranode even though the structural integrity of axoglial junctions was intact (*Zhang et al., 2013*). βII spectrin contributes to the sub-membranous cytoskeleton of the axon-linking membrane proteins to actin and appears to act as a barrier limiting the lateral diffusion of membrane proteins. We found that paranodal expression of βII spectrin was reduced following axotomy and a reduction in this sub-membranous barrier is compatible with the lateral movement of Kv1 channels into this region that we observed. As well as Kv1 channels we also see caspr2 overlapping with paranodal markers following nerve injury and again such paranodal localisation of caspr2 was also reported in mice in which axonal βII spectrin is conditionally ablated. Caspr2 complexes with and is important for the correct localisation of Kv1 channels (*Poliak et al., 1999*; *2003*) suggesting that this whole protein complex is mis-localised following nerve injury. Although loss of a sub-membranous barrier to diffusion of the VGKC-complex is one explanation for their paranodal localisation, we do not yet have a full understanding of the regulation of Kv1 channels trafficking. Phosphorylation events may have a role to play as Kv1.2 can undergo phosphorylation, which impacts on surface expression/localisation (*Gu and Gu, 2011*; *Yang et al., 2007*).

## The effect of altered Kv1 channels subunit composition and localisation on axonal excitability and neuropathic pain

Following axonal injury sensory axons become hyper-excitable and this is important in driving and maintaining neuropathic pain (*Han et al., 2000*). A recent study showed that myelinated sensory fibres are key in maintaining mechanical allodynia in several neuropathic pain models (*Xu et al., 2015*). Spontaneous activity and mechanical stimulus evoked activity has been recorded in myelinated afferents innervating neuroma using microneurography (*Nyström and Hagbarth, 1981*). The role of Kv1 channels has mainly focussed on their importance in suppressing excitability at the soma rather than the axon following injury. The expression of a number of α Kv1 channels sub-units has been documented to decrease following peripheral axotomy including Kv1.1, 1.2 (*Everill et al., 1998*; *Hao et al., 2013*; *Ishikawa et al., 1999*; *Kim et al., 2002*; *Yang et al., 2004*) and in some reports Kv1.4 (we did not see a reduction in Kv1.4 using western blot analysis of DRG lysate following sciatic axotomy, however, this is a less proximal lesion compared to spinal nerve ligation [*Everill et al., 1998*]). Correspondingly, the K currents mediated by such channels are reduced when measured at the soma (*Yang et al., 2004*) both in small and large diameter DRG cells. The focus has therefore been on the loss of K currents within the soma which normally act as a 'break' on excitability, and combined with increased excitatory drive for instance due to the dysfunction of voltage-gated sodium channels and hyperpolarization-activated cyclic nucleotide-gated (HCN) channels this leads to ectopic activity (*Waxman and Zamponi, 2014*). While changes at the soma are undoubtedly important, ectopic impulses also arise along the axon. There has been much less focus on the distribution and function of Kv1 channels within the axon following peripheral nerve injury.

The function of Kv1 channels is critically dependent on their targeting to specific neuronal compartments (*Trimmer, 2015*). The expression of Kv1 channels within the DRG soma and the axon should not be assumed to be the same: For instance the expression of Kv1.6 remains stable within the DRG both at the level of mRNA and protein (see *Kim et al. (2002)*, *Yang et al. (2004)* and our own data) but as we show here expression markedly increases in axons within peripheral nerve and dorsal root. Altered α subunit composition and localisation of Kv1 channels is likely to have functional implications. In the naïve state, the Kv1 channels Kv1.1 and 1.2 are located within the juxtaparanode, below the insulating myelin sheath and at least in peripheral nerves have little functional impact on nodal excitability and conduction (*Poliak et al., 2003*; *Chiu and Ritchie, 1980*; *Sherratt et al., 1980*; *Rasband et al., 1998*). During development when Kv1 channels are observed

in the node and paranode using specific blockers of this K current suggests that these Kv1 channels prevent re-entrant excitation in motor axons (*Vabnick et al., 1999*). Following primary demyelination, the re-distribution of Kv1 channels to the paranode acts to suppress continuous conduction in demyelinated axons (*Rasband et al., 1998*). In certain contexts therefore and especially when Kv1 channels begin to encroach on the paranodal regions there is evidence that these channels can suppress excitability of the axon. As has been previously demonstrated we have found that the rate of ectopic activity within myelinated axons is very high in the first week and then decreases the longer the time elapsed following the initial injury (*Campbell et al., 1988*). Understanding adaptive mechanisms to suppress such hyper-excitability will potentially provide insight as to why in certain patients such mechanisms fail leading to chronic pain states. Over a similar time period as this reduction in spontaneous activity we have observed increased expression of Kv1.4 and 1.6 as well as redistribution of Kv1 channels to the paranode and juxtaparanode domains. Selective inhibition of Kv1 channels with α-DTX reinstates a higher level of ectopic activity, increases mechanosensitivity of afferents innervating the neuroma and on behavioural testing also exacerbates mechanical hypersensitivity, which had begun to normalise at 3 weeks post injury. α-DTX blocks Kv1.1, Kv1.2 and Kv1.6. As expression of Kv1.1 and 1.2 are decreased while Kv1.6 is increased, most probably the effect seen with the toxin is through Kv1.6. Selective inhibition of Kv1.4 did not recapitulate these events emphasising the role of Kv1.6 (and subunits with which it complexes) in suppressing hyperexcitability. Neuronal Kv1 proteins form heterotetramerization of α subunits, which also associate with auxiliary Kvβ subunits (*Jan and Jan, 2012*), adding complexity in ascribing function to individual α subunits. α subunits confer particular pharmacological and biophysical properties on these channels and in addition there may be interactions between subunits. For instance Kv1.4, usually shows N-type rapid inactivation through an N-terminal inactivation ball however this can be over-ridden if associated with a Kv1.6 α subunit (*Roeper et al., 1998*), due to its NIP (N-type inactivation prevention) domain.

In conclusion, we have found major changes in Kv1 channels subunit composition and distribution within the axolemma of myelinated axons following traumatic nerve injury. In contrast to the soma in which Kv1 channels expression is reduced this increased availability of Kv1 channels within the paranodes and altered subunit composition appears to fulfil an adaptive role in suppressing excessive excitability in myelinated afferents.

## Materials and methods

### Animals and surgery

Adult male Sprague-Dawley rats were used in accordance with UK Home Office and Pontificia Universidad Catolica's regulations (animals in the UK were purchased from Charles-River UK, animals from Chile were purchased onsite). Rats were group housed and placed on a 12 hr-light 12 hr-dark cycles. Two different nerve injury models were used: the neuroma model and the L5 spinal nerve transection (SNT) model. The neuroma model of neuropathic pain was based on the TNT model (*Dorsi et al., 2008*), but performed with some modifications. Briefly, the sciatic nerve was dissected free of adjacent tissue, ligated with a suture, and cut proximal to its bifurcation. The needle from the suture was passed through a subcutaneous tunnel to the lateral aspect of the hindlimb where it was pushed through the skin. The nerve was drawn into the tunnel until the ligature is adjacent to the skin. The suture was cut, and the incision closed. The suture tied to the distal end of the sciatic nerve was visible through the skin and served as the target for mechanical stimuli. An analogous site served as the target on the contralateral hindlimb. For the L5 SNT model (*Kim and Chung, 1992*), one-third of the L6 transverse process was removed and the L5 spinal nerve was identified and dissected free from the adjacent L4 spinal nerve and then tightly ligated using 6–0 silk and then transected distally to the suture. Sham-operated animals served as a control. We used these two different models as the neuroma model is the most adequate for performing behavioural tests as the injured nerve can be directly stimulated, while the L5 SNT model has the advantage that it gives certainty that all the dorsal root axons studied had their peripheral terminals injured. For both models animals were deeply anaesthetised with a mix of isofluorane and oxygen. Postoperative analgesia was given for the first 5 days postop (tramadol 50 mg/kg/day p.o). Animals were checked every day after surgery to check for self-mutilation behaviour (autotomy), which prompted us to sacrifice the

**Table 1.** Different antibodies used in the study.

| Antibody | Concentration used IHC WB | | Company |
|---|---|---|---|
| Rabbit anti Pan voltage gated sodium channel (Cat No. S6936) | 1:1000 | | Sigma-Aldrich |
| Mouse anti Kv1.2 (K14/16.2) | 1:100 | 1:500 | UC Davis/NIH NeuroMab Facility |
| Mouse anti Kv1.1 (K36/15.1) | 1:100 | 1:200 | UC Davis/NIH NeuroMab Facility |
| Mouse anti Kv1.4 (K13/31) | 1:100 | 1:200 | UC Davis/NIH NeuroMab Facility |
| Mouse anti Kv1.6 (K19/36) | 1:100 | 1:500 | UC Davis/NIH NeuroMab Facility |
| Guinea Pig anti Caspr | 1:1000 | 1:1000 | From Dr Manzoor Bhat - UT Health Science Center San Antonio (*Bhat et al., 2001*) |
| Rabbit anti Caspr2 (ab105581) | 1:500 | 1:400 | Abcam |
| Rabbit anti Pan Neurofascin | 1:500 | | Gift from Prof Peter Brophy- University of Edinburgh (*Pomicter et al., 2010*) |
| Mouse anti βII spectrin (Clone 42) | 1:500 | 1:1000 | BD Bioscience |
| GAPDH | 1:10000 | | Abcam |
| PGP 9.5 | 1:5000 | | Ultraclone |

IHC: Immunohistochemistry; WB: Western Blot analysis.

animal. Calculation of the sample size needed was done for each experiment as described below. Experimental protocols were reviewed and approved by 'Coordinación de Ética, Bioética y Seguridad de las investigaciones UC' (experiments done in Chile) and were performed in accordance with the UK Home Office regulations (experiments done in the UK). We report this study in compliance with the ARRIVE guidelines (*Kilkenny et al., 2010*) (20 points checklist).

## Patients and controls

The study was conducted at Hospital Clinico UC-Christus in Santiago, Chile. Morton's neuroma patients that were due surgery for resection of painful neuromas were recruited for donating a small sample of the tissue resected during surgery. Control samples were obtained from subjects undergoing hand reconstructive surgery in where the sural nerve is harvested and used as a bridge to connect disrupted ends of motor nerves in the hand. A small sample for these healthy sural nerves was collected to use as control in this study. We used the Numeric Rating Scale (NRS; which is a self-reporting scale where 0 is no pain and 10 is the worst imaginable pain) to assess for pain before surgery. Informed consent was obtained from all subjects before surgery. The study protocol was assessed and approved by the Ethics Scientific Committee of the School of Medicine Pontificia Universidad Catolica de Chile (reference number 14–389). The sample sizes were calculated using a power of 80% and an α error of 0.05%, assuming a 2 times increase or decrease in Kv channel expression with a variance of 0.6 from the mean, which resulted in a sample needed of 3 patients per group.

## Histology

After a defined survival time (7 and 21 days), animals were terminally anaesthetized with pentobarbital and transcardially perfused with 0.9% heparinised saline. The L5 DRG, the L5 spinal nerve, and the sciatic nerve were removed. We dissected the sciatic nerve free from connective tissue and collected 5 mm from the site of the neuroma and 5 mm from a site 1 cm proximal. Tissue for immunohistochemistry was post fixed in 4% paraformaldehyde (PFA) in 0.1 M phosphate buffer (PB) for 30 min and cryoprotected in 20% sucrose for 3 days. Tissue obtained from patients was fixed immediately after resection in 4% PFA for 30 min and then cryoprotected in 20% sucrose for 3 days. The samples were embedded in OCT, cryostat cut (8 µm) and thaw-mounted onto glass slides. Sections were pre-incubated in buffer (PBS, pH 7.4, containing 0.2% Triton X-100 and 0.1% sodium azide)

containing 10% normal donkey serum for 30 min and then incubated with primary antibodies over-night at room temperature. Primary antibodies used are shown in *Table 1*. Following primary anti-body incubation, sections were washed and incubated for 2 hr with secondary antibody solution (donkey anti-rabbit Cy3 1:400; goat anti guinea pig AMCA 1:100, donkey anti mouse FITC 1:400; all from Stratech, UK). Slides were washed with PBS, cover-slipped with Vectashield mounting medium (Vector Laboratories, UK) and visualised under a Zeiss Axioplan 2 fluorescent microscope (Zeiss, UK). All quantification of different IHC parameters was done with the investigator blinded to the identity of the group to which the animals belonged. Nodal quantification was done by assessing on average 31 nodes per animal, and using 4–5 animals per condition. For quantification of βII spectrin the inten-sity of the immunofluorescence of the axonal paranodal area (identified by caspr staining) was mea-sured and the background of each section was subtracted. Then, measurements were normalised against the mean of the controls (naïve axons). The sample sizes were calculated using a power of 80% and an $\alpha$ error of 0.05%, assuming a 2 times increase or decrease in expression with a variance of 0.5 from the mean, which resulted in a sample needed of 4 animals per group.

We quantified sodium channel clusters following the following criteria: (1) typical nodes were nodes where the Nav channels fill the gap at the node of Ranvier as identified by the paranodal staining of caspr on both sides of the node, (2) split nodes were nodes that had two distinct Nav channels accumulations, separated by a gap in the Nav channel staining within the same fibre and with each Nav channels accumulation flanked on one side with caspr staining, or (3) heminodes were nodes where the caspr staining located on only one side of a contiguous Nav channel accumulation, (4) while those Nav channel accumulations lacked an association with caspr were classified as 'naked' accumulations.

## Western blot analysis

Tissue was collected, quickly frozen in liquid nitrogen and was homogenized in NP40 lysis buffer (20 mM Tris, pH 8, 137 mM NaCl, 10% glycerol, 1% NP-40, 2 mM EDTA), 20 μM leupeptin, 5 mM sodium fluoride, 1 mM sodium orthovanadate, 1 mM PMSF and protease inhibitor cocktail). The lysate were spun at 13,000 rpm at 4°C for 15 min and the protein concentration of supernatant was determined using a BCA Protein Assay kit (Thermo Scientific). 50 μg of each sample was separated using 8% or 10% SDS-PAGE, and transferred to nitrocellulose membranes. Membranes were then blocked in 10% skimmed milk in PBS-T (PBS+ 0.1% Tween 20) for 1 hr at room temperature. Mem-branes were incubated with primary antibody (anti mouse Kv1.1, Kv1.2, Kv1.4, Kv1.6, GAPDH, PGP9.5 and anti-rabbit Caspr2 as shown in *Table 1*), diluted in PBS-T at 4°C overnight. After washing with PBS-T for 6 times and5 min each time, membranes were incubated with sheep anti-mouse or donkey anti rabbit HRP-conjugated secondary antibody (1:10,000–1:20,000; ECL, GE Healthcare, Amersham, UK) at room temperature for 1 hr. After several PBS-T washes as described above mem-branes were revealed using ECL-prime reagent (GE Healthcare) for 5 min for detection by autoradiography.

For WB of Kv1.4 and Kv1.6 in rat tissue (sciatic nerve and DRG) the membranes were cut in three pieces; the top piece was probed with Kv1.4 antibody, the middle one was probed with Kv1.6 anti-body and the bottom one probed with GAPDH antibody. For WB of Kv1.1 and for Kv1.2 the mem-branes were cut in 2 pieces: the top one was probed with either Kv1.1 or Kv1.2 antibody and the bottom one was probed with GAPDH antibody. The 2 or 3 pieces of the membranes were lined up as a single membrane before exposing it to the film so that the molecular weight can be calculated by measuring the running distance of the molecular weight marker and the target bands. This could be done as the bands labelled by the antibodies have quite different molecular weights. This allowed us to optimize the use of the tissue obtained from animals and reduce the number of ani-mals needed (in accordance with our obligations under animal licensing procedures).

## Quantification and analysis

For Western Blots analysis, films were scanned with Cannon Scanner (LiDE 210), and the intensity of specific bands was quantified using Quantity One software (Bio-Rad). The same size rectangle was drawn around each band to measure intensity, and the background was subtracted. Target band detected was normalized against loading control GAPDH or PGP9.5 correspondingly for analysis. The sample sizes were calculated using a power of 80% and an $\alpha$ error of 0.05%, assuming a 2 times

increase or decrease in expression with a variance of 0.5 from the mean, which resulted in a sample needed of 4 animals per group (we used 6 animals per group in case we had to put any animal down due to autotomy).

## Electron microscopy

Sciatic nerves were dissected at the site of the neuroma and were processed for resin embedding as previously described (Huang et al). Briefly nerves were post fixed in 3% glutaraldehyde at 4°C overnight, washed in 0.1 M PB, osmicated, dehydrated, and embedded in epoxy resin (TAAB Embedding Materials, UK). Longitudinal sections 1 μm thick were cut on a microtome and stained with toludine blue before being examined on a light microscope. Ultrathin sections were cut on an ultramicrotome and stained with lead uranyl acetate. Sections were mounted on unsupported 100 mesh grids. Sections were visualised on a PHILIPS TECNAI 12 BIOTWIN transmission electron microscope at the Unidad de microscopia avanzada, Pontificia Universidad Catolica de Chile. We measured the diameter of the axons at the site of the node, the maximal and minimal distance between interloops, the distance between the glia and the axon, the number of detached loops, and the number of everted loops using Image J (NIH, USA) and a 135000x magnification. We quantified between 8 and 14 nodes per animal, and we used 5 animals per condition (sample sizes were calculated using a power of 80% and an α error of 0.05%, assuming a change in distance of 50% with a variance of 0.4, which resulted in a sample needed of 4 animals per group, however due to the difficulty in the technique we included one more animals in each group). The investigator was blinded to the treatment group of each specimen, however, this was sometimes difficult to conceal as the anatomy in the injured nerves was much more disrupted than in naive nerves.

## In vivo electrophysiological recording

Recordings were performed under anaesthesia (urethane, 1.5 g/Kg, i.p.) on naïve rats (n = 10, 222 neurons), or after sciatic nerve ligation at 2 days (n = 4, 291 neurons), at 7 days (n = 7, 241 neurons), and at 21 days (n = 7, 237 neurons). A tracheotomy was performed and the L5 dorsal roots and DRGs were exposed via laminectomy. Sciatic nerve neuroma with proximal nerve (5–6 mm long) and contralateral uninjured sciatic nerve were exposed. The contralateral sciatic nerve was acutely cut to disconnect from the periphery just before recording. The entire site was covered in agarose gel and four chambers created by removing blocks of this gel. These were 1) neuroma chamber, containing ipsilateral neuroma and part of sciatic nerve which is subjected to stimulation; 2) acutely cut nerve end chamber, containing contralateral sciatic nerve proximal end; 3) DRGs chamber, containing L5 DRGs from both sides; 4) spinal recording chamber, containing part of L5 dorsal roots from both sides near entry zone to spinal cord. The neuroma chamber and nerve cut end chamber were filled with mineral oil during stimulation, and the oil was replaced with αDTx (100 nM in saline) during toxin application. The DRGs chamber was filled with saline or αDTx/saline solution, and the recording chamber was always filled with mineral oil. The pool temperatures were not controlled, but as animals were warmed using an infrared lamp from the back, the pool was therefore heated, and typically was at 34–35°C. Just before recording, the L5 dorsal root was cut near entry zone, a filament was teased out and hooked up for recording. Each filament was stimulated electrically with increasing current to recruit sequentially each conducting axon in that filament. The conduction velocity of each conducting axon was noted. Thus, the number of functioning axons in each filament was determined (typically, 6–10). Spike discrimination was used to detect the number different axons firing spontaneously in each filament (typically 0–3) during a pre-treatment baseline and under 3 different treatment conditions: 1) no αDTx in any of the chambers; 2) αDTx in neuroma or nerve cut end chamber; 3) αDTx in neuroma or nerve cut end chamber and DRGs chambers. The αDTx was applied for at least 20 min before recording. An independent investigator prepared the drugs individually and labelled them for each animal according to the randomization schedule. Data analysts were blinded as the conditions under which all recording were made.

Signals were amplified with an AC-coupled amplifier (Neurolog NL104A with headstage NL100AK), then high-pass- and low-pass filtered (Neurolog NL125) at 500 Hz and 5 KHz frequencies. The filtered signals were passed through a Humbug 50 Hz noise eliminator (Quest scientific, Vancouver, BC, Canada), further amplified (Neurolog NL 106), fed to an analog-to-digital converter PowerLab, and sampled at 20 KHz with Labchart software (ADinstruments, UK). Stimulation (200 μs

square-wave pulses) was delivered from a stimulus isolator (Neurolog, NL800A). The filter settings used strongly favours recordings from A-fibres and not C-fibres. All the fibres recorded to nerve stimulation conducted in the A fibre range (>2 m/sec). The size of the filaments recorded was also unfavourable for detecting clear single unit C fibre activity.

Three minutes baseline was recorded to examine spontaneous activity. The percentage of spontaneously firing units was calculated as the number of spontaneously active units divided by the number of conducting fibres determined in recruitment recording. The firing rate was calculated as the total number of spikes during recording divided by the time recorded. The mean firing rate per unit was the firing rate divided by the total number of different units recorded in each treatment group.

For the axonal mechanosensitivity experiments (n = 4, 259 neurons), mechanical stimulation was applied to the neuroma using increasing forces of von Frey filaments (4, 8, and 15 g), and the number of distinct spikes (neurons) firing in response were counted following spike discrimination. The total number of conducting axons in each filament was determined by incremental electrical stimulation of the sciatic nerve. The percentage of mechanosensitive units was calculated as the number of different neurons responding by firing action potentials upon mechanical stimulation, divided by the number of conducting fibres (which was determined in the same way as for the spontaneous activity experiments). Axonal mechanosensitivity was assessed before and after toxin application at 21 days after axotomy. Mechanosensitivity experiments were carried out on separate animals to spontaneous activity experiments to ensure that any spontaneous activity encountered was not caused acutely by the repeated mechanical stimulation of the neuroma.

Data was analyzed using software Labchart. Statistics comparing proportions of neurons exhibiting either spontaneous activity or mechanosensitivity were performed using chi-square test with Yates correction. Values were reported as percentages, calculated from the proportions.

## Assessments of mechanical sensitivity

Mechanical withdrawal thresholds were assessed by applying a range of Von Frey hairs (Somedic, Sweden) to the skin over the neuroma site (labelled with a suture as previously described). Animals were randomised to receive either subcutaneous αDTX (0.5 ml at 100 nM in saline, Alomone, UK) or saline (which was administered locally at the site of the neuroma) using a computer-generated random sequence. The sample sizes were calculated using a power of 80% and an α error of 0.05%, assuming a 60% decrease in withdrawal threshold with a variance of 25% from the mean, which resulted in a sample needed of 7 animals per group. Experimental groups were the following: baseline with vehicle, baseline with toxin; day 3 after surgery with vehicle, day 3 with toxin; day 7 with vehicle, day 7 with toxin; day 21 with vehicle, day 21 with toxin. To reduce the amount of animals of the study the animals that received saline only were used again for the consecutive time-points. The animals that received toxin had to be sacrificed after testing, as the toxin is irreversible. The toxin or saline were injected 30 min before testing. Autotomy after nerve injury (especially neuroma model) appears at around 10 days after injury. Therefore, we allocated 2 extra animals for the saline group, and 2 extra for toxin day 21. We had to sacrifice 1 animal from the saline group at day 6, and 2 animals from the toxin group day 21 (at day 15 and 17 after injury respectively), due to self-mutilating behaviour. For testing, rats were gently restrained using a towel on a table. Calibrated von Frey hairs were applied to the skin covering the neuroma until the fibre bent. Withdrawal of the limb by the animal was recorded as a response. The 50% withdrawal threshold was determined using the up-down method (Dixon, 1980). An independent investigator prepared the drugs individually and labelled them for each animal according to the randomization schedule. Operators and data analysts were blinded throughout the study. The data were distributed normally and the differences between groups was analysed using a 2 way ANOVA repeated measures. Values were reported as mean ± SEM.

This experiment was repeated for testing CP339818 (Kv1.4 blocker; #C-115, 0.5 ml at 300 nM in saline Alomone Labs UK). We randomly allocated 8 animals per group, and we had to put one animal from the saline group down due to autotomy at day 12.

## Acknowledgements

DLHB is a Welcome senior clinical scientist (ref. no. 095698z/11/z). MC received Conicyt PAI Apoyo al Retorno del investigador en el extranjero (Folio 82130016), New Faculty award from Pontificia

Universidad Catolica (2755010), and funds from Núcleo Milenio RC120003 to complete this work. The work was supported in part by a Senior Investigator award to SBM from the Wellcome Trust (ref 97903). We would like to thank Prof Peter Brophy of the University of Edinburgh for the gift of the pan-neurofascin antibody.

## Additional information

### Funding

| Funder | Grant reference number | Author |
| --- | --- | --- |
| Pontificia Universidad Catolica de Chile | New Faculty Award 2755-010-81 | Margarita Calvo |
| Comisión Nacional de Investigación Científica y Tecnológica | Apoyo al Retorno del investigador en el extranjero Folio 82130016 | Margarita Calvo |
| Wellcome Trust | Wellcome strategic award | Stephen B McMahon David LH Bennett |
| Wellcome Trust | Senior Wellcome Clinical Scientist (ref. no. 095698z/11/z) | David LH Bennett |
| Comisión Nacional de Investigación Científica y Tecnológica | FONDAP-15150012, Ministerio de Economia, Millennium Nucleus-P-07-011-F | Felipe A Court |

The funders had no role in study design, data collection and interpretation, or the decision to submit the work for publication.

### Author contributions

MC, NR, ABS, AB, LZ, FAC, SBM, DLHB, Conception and design, Acquisition of data, Analysis and interpretation of data, Drafting or revising the article; DI, NZ, Acquisition of data, Analysis and interpretation of data; PA, Collected tissue from patients, Acquisition of data, Contributed unpublished essential data or reagents; MAB, Conception and design, Analysis and interpretation of data, Drafting or revising the article, Contributed unpublished essential data or reagents

### Author ORCIDs

Margarita Calvo, http://orcid.org/0000-0003-3349-9189

### Ethics

Human subjects: Informed consent, and consent to publish, was obtained from all subjects to collect and analyze nerve samples before surgery. Subjects underwent surgery by indication of their physician and samples were obtained from biological tissue that was otherwise due to be incinerated. The study protocol was assessed and approved by the Ethics Scientific Committee of the School of Medicine Pontificia Universidad Catolica de Chile (reference number 14-389).

Animal experimentation: This study was performed in strict accordance with UK Home Office and Pontificia Universidad Catolica's regulations. Experimental protocols were reviewed and approved by "Coordinación de Ética, Bioética y Seguridad de las investigaciones UC" (experiments done in Chile- Protocol CBB230/2013) and were performed in accordance to the UK Home Office regulations (experiments done in the UK). We report this study in compliance with the ARRIVE guidelines (20 points checklist).

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
