## [Decision Letter]

Thank you for submitting your work entitled "Changes of shaker-type Kv channels in the nodal complex suppress hyperexcitability in neuropathic pain" for consideration by *eLife*. Your article has been favorably evaluated by three peer reviewers, including Félix Viana and Peggy Mason, who is a member of our Board of Reviewing Editors. The evaluation was overseen by Peggy Mason as the Reviewing Editor and Gary Westbrook as the Senior Editor.

The reviewers have discussed the reviews with one another and the Reviewing Editor has drafted this decision to help you prepare a revised submission.

Summary:

The main finding of this work is a downregulation of Kv1.1 and Kv1.2 at the juxtaparanode, followed by a delayed "compensation" of Kv1.4 and Kv1.6 expression that also spreads into the paranode. These changes correlate with disruption of the cytoskeletal protein βII spectrin, a protein involved in Kv channel clustering at the juxtaparanode. The functional consequence of the Kv1 dynamic changes is a transient increase in spontaneous activity and mechanosensitivity.

Essential revisions:

Figure 2: it is curious that the authors show in Figure 1 that there is a loss of Kv1.2 from the neuroma, but then show Kv1.2 redistribution in the neuroma in Figure 2 recognize that these are from very different time points, with the later time point being shown first. This is somewhat confusing. I suspect that the redistribution of the channels reflects a damaged paranodal junction (see comment for Figure 3). This is also likely highly dependent on how close to the site of injury the analysis was performed. Was there a difference in the degree of Kv1 channel 'invasion' to paranodes depending on where in the nerve one looked, (i.e. close to the neuroma or far from the neuroma)?

Figure 6: the authors are commended for attempting to use human samples to compare with what was obtained in the rodent model. However, the staining in Panel 6a is not convincing. Neither the Kv1.4 nor the Kv1.6 staining is convincing. Lower magnification images should be shown for the Kv1.6 – it looks like the immunostaining extends beyond where the juxtaparanode should end. The Kv1.4 staining is simply not compelling.

How can KV1.2 be used as a marker of the paranoid when its distribution is the object of inquiry? Why not use caspr2?

---

## [Author Response]

Essential revisions:

*Figure 2: it is curious that the authors show in Figure 1 that there is a loss of Kv1.2 from the neuroma, but then show Kv1.2 redistribution in the neuroma in Figure 2 recognize that these are from very different time points, with the later time point being shown first. This is somewhat confusing. I suspect that the redistribution of the channels reflects a damaged paranodal junction (see comment for Figure 3). This is also likely highly dependent on how close to the site of injury the analysis was performed. Was there a difference in the degree of Kv1 channel 'invasion' to paranodes depending on where in the nerve one looked, (i.e. close to the neuroma or far from the neuroma)?*

Thanks for the comment, as this is a very good point. To answer this question we repeated the experiment (i.e. neuroma at day 7) and collected nerve from the injury site close to neuroma) and from a site 1cm proximal to the injury site (far from neuroma). For Kv1.2 we found that the results were the same when we looked at a site close to the neuroma and at a site far from the neuroma with the movement of Kv1.2 into the paranodal region in both situations. In comparing the neuroma site and 1cm proximal to the neuroma there was no difference in the distance measured from the Nav staining to the end of Caspr staining (p = 0.8), no difference in the distance measured from the Nav staining to the start of Kv1.2 staining (p = 0.9), and in the difference between these measurements (p = 0.6). Regarding caspr2, we also found that it moves to the paranode at both sites, close and far from the neuroma. Comparing these two sites there was no difference in the distance measured from the Nav staining to the end of Caspr staining (p = 0.9), no difference in the distance measured from the Nav staining to the start of caspr2 staining (p=0.1), and no difference (p=0.06) in the difference between these measurements. We have now added these new results to the manuscript (Results and Discussion) and in Figure 3 (former Figure 2).

*Figure 6: the authors are commended for attempting to use human samples to compare with what was obtained in the rodent model. However, the staining in Panel 6a is not convincing. Neither the Kv1.4 nor the Kv1.6 staining is convincing. Lower magnification images should be shown for the Kv1.6 – it looks like the immunostaining extends beyond where the juxtaparanode should end. The Kv1.4 staining is simply not compelling.*

The reviewer is right that the staining we showed for Kv1.6 and for Kv1.4 in human tissue is not as optimum as that in preclinical animal models. Therefore, we changed those two images for other examples. We should say, that this staining is extremely difficult to do in human nerves (due to the fact that for instance we have limited control as to the duration of time before the nerve is placed in fix following surgery), and after several optimizing steps this is our best attempt. We added high magnification pictures in the figure to fit the style, but in Figure 12 we show low magnification of the same images (both are in Morton neuroma). There is a higher background than we would like but we do think that there is juxtaparanodal staining.

Author response image 1.**DOI:**
http://dx.doi.org/10.7554/eLife.12661.025

How can KV1.2 be used as a marker of the paranoid when its distribution is the object of inquiry? Why not use Caspr2?

We assume that this is a typo and this question relates to using Kv1.2 as a marker of the juxtaparanode (not paranode). Assuming this is the case, we agree with the reviewer that it is best not to refer to Kv1.2 as a marker of the juxtaparanode. It is the subject of inquiry and is a protein normally localized to the juxtaparanode and it is re-distributed after injury. What we have done is measure the distance between the voltage gated sodium channel accumulation (Nav) and the end of caspr (which we used as a marker of the paranode and importantly this distance doesn’t change with nerve injury). We also measured the distance between Nav and the start of the Kv1 channels. We then obtained the difference between these two distances. If the distance was a positive value it meant that there is a gap, albeit very small, between this paranodal protein and the Kv1 channels. If this distance was a negative value it meant that the Kv1 channels have moved into the paranode and co-localise with caspr (see Figure 13). We did these same measurements for caspr2 another protein normally localized to the juxtaparanode (Figure 3), which is also re-distributed along the Kv1 channels.

Author response image 2.**DOI:**
http://dx.doi.org/10.7554/eLife.12661.026

To make this clearer we now included explicit references to these measurements (Nav to caspr and Nav to Kv1/caspr2). Note that caspr is not re-distributed after nerve injury.

We have re-labeled the charts in Figure 3 and Figure 8 (former Figure 2 and Figure 7) to avoid saying “end of PRN and start of JXP” which we agree is confusing and not the most appropriate terms.